# 3D printable strong and tough composite organo-hydrogels inspired by natural hierarchical composite design principles

Quyang Liu[1], Xinyu Dong[1], Haobo Qi[1], Haoqi Zhang[1], Tian Li[1], Yijing Zhao[1], Guanjin Li[1] & Wei Zhai ®[1] ✉

Fabrication of composite hydrogels can effectively enhance the mechanical and functional properties of conventional hydrogels. While ceramic reinforcement is common in many hard biological tissues, ceramic-reinforced hydrogels lack a similar natural prototype for bioinspiration. This raises a key question: How can we still attain bioinspired mechanical mechanisms in composite hydrogels without mimicking a specific composition and structure? Abstracting the hierarchical composite design principles of natural materials, this study proposes a hierarchical fabrication strategy for ceramic-reinforced organo-hydrogels, featuring (1) aligned ceramic platelets through direct-ink-write printing, (2) poly(vinyl alcohol) organo-hydrogel matrix reinforced by solution substitution, and (3) silane-treated platelet-matrix interfaces. Unit filaments are further printed into a selection of bioinspired macro-architectures, leading to high stiffness, strength, and toughness (fracture energy up to 31.1 kJ/m$^2$), achieved through synergistic multi-scale energy dissipation. The materials also exhibit wide operation tolerance and electrical conductivity for flexible electronics in mechanically demanding conditions. Hence, this study demonstrates a model strategy that extends the fundamental design principles of natural materials to fabricate composite hydrogels with synergistic mechanical and functional enhancement.

Hydrogels are three-dimensional polymeric networks that can retain large amounts of water, but conventional hydrogels with limited cross-linking and loose polymer networks are relatively weak and fragile to meet the demands of real-life applications[1]. To improve their mechanical properties such as strength and toughness, composite hydrogels are formulated by introducing micro- and nano-fillers, including MXene[2–4], graphene oxide[5,6], carbon nanotubes (CNTs)[7], and polymer fibers[8,9]. Ceramics are also an easily accessible and cost-effective reinforcement material with excellent stiffness, strength, and chemical stability. Indeed, nature presents abundant examples of organic-inorganic composite materials, such as nacre[10], bone[11], and glass sponge (*Euplectella aspergillum*)[12]. They are reinforced by aligned mineral platelets embedded in a soft matrix. While the mineral phase provides high stiffness and strength, the soft matrix in between them facilitates load transmission and energy dissipation. This combination results in stiffness similar to that of the mineral constituent but toughness orders of magnitude higher[13]. That being said, these materials are typically hard biological tissues with a predominant mineral content, and their mechanical properties differ from those of soft materials that are elastic and flexible like hydrogels. Despite the apparent contrast in compositions, an intriguing question arises: Can the fundamental design principles of these hard biological tissues still be extracted and applied to strengthen and toughen composite hydrogels?

The excellent stiffness, strength, and toughness of these natural materials result from their intricate hierarchical organic-inorganic

[1]Department of Mechanical Engineering, National University of Singapore, 9 Engineering Drive 1, Singapore, Singapore. ✉e-mail: mpezwei@nus.edu.sg

composite structure. Firstly, they feature a combination of (1) aligned stiff anisotropic particles or fibers with a thickness ranging from one to a few hundred nanometers, (2) embedded in a soft and tough matrix, with (3) a tight interface between the stiff elements and the soft matrix[14]. Moreover, natural materials form through bottom-up growth, where they are assembled by the most fundamental building blocks in a hierarchical manner. For example, the hierarchical structure of bone has up to seven levels of organization, constructed by hydroxyapatite nanocrystals and collagen molecules as basic building blocks[15]. This hierarchical organization, from molecular to macro scales, achieves a strong coupling between the organic and inorganic constituents, fostering synergistic interactions across multiple length scales.

While reproducing the exact bottom-up growth and complex hierarchical structures of natural materials can be a tantalizing yet notably challenging endeavor, applying their design principles through practical engineering methods may offer a more feasible route to fabricating bioinspired composite hydrogels. To this end, various structural engineering approaches have been developed to fabricate composite hydrogels with anisotropically aligned microstructures[16], such as using magnetic[5,17,18] and electric fields[7,19], mechanical training[20,21], freeze casting[22,23], and self-assembly[24]. Recent advances in direct-ink-write (DIW) 3D printing also showcase its capability to align anisotropic particles along printed filaments by the extrusion shear force[25–30]. Moreover, DIW 3D printing offers precise material deposition as per computer-aided designs at the macro scale, which, in combination with the shear-induced alignment at the micro scale, opens more possibilities for fabricating hierarchically structured composite materials. Meanwhile, hydrogels can be reinforced by material engineering approaches that enhance their macroscopic mechanical properties through structural changes at molecular and nano scales, such as introducing salt ions[1], inducing hydrophobic aggregation[23], and constructing nanocrystalline domains[31]. Whereas these structural and material approaches primarily work on specific length scales, a fabrication strategy integrating both structural and material engineering is needed to implement the hierarchical design principles of natural materials in composite hydrogels.

Herein, this study presents a hierarchical fabrication strategy leveraging the fundamental design principles of natural materials to achieve strong and tough ceramic-reinforced organo-hydrogels. The composite organo-hydrogels consist of (1) ceramic platelets of around 200 nm thickness aligned by DIW 3D printing, (2) a highly crystalline poly(vinyl alcohol) (PVA) organo-hydrogel matrix reinforced by solution substitution, and (3) enhanced ceramic-polymer interfaces by silane surface treatment on the ceramic platelets. Filaments with aligned ceramic platelets, which serve as the basic building block, are further 3D printed into a selection of bioinspired macro-architectures. By our hierarchical fabrication strategy, the composite organo-hydrogels exhibit a combination of high stiffness, strength, and toughness, achieved via effective toughening mechanisms inspired by natural materials and synergistic multi-scale energy dissipation. Also, considering a lack of inherent functionality in both ceramic platelets and PVA matrix, our strategy simultaneously endows the composite organo-hydrogels with excellent operation tolerance by substitution in a glycerol-water solution, and electrical conductivity by Mg-thermic reduction of ceramic platelets and incorporation of ferric ions in the substituting solution. With both enhanced mechanical and functional properties, the composite organo-hydrogels can be applied in various mechanically demanding applications, including flexible electronics as demonstrated in this study.

## Results
### Fabrication and properties of composite organo-hydrogel filaments

Figure 1a illustrates the fabrication process of composite organo-hydrogels. The process started with modification of the ceramic fillers,

where titania-coated alumina platelets were treated by a magnesium-thermic reduction process to create oxygen vacancies in the ceramic oxides and endow them with electrical conductivity[29,32]. As revealed by X-ray photoelectron spectroscopy (XPS), Mg-thermic reduction occurred at the titania coating of the platelets, leading to formation of $Ti^{3+}$ (with peaks at 456.5 eV and 461.6 eV) in addition to $Ti^{4+}$ (at 458.5 eV and 464.2 eV) in the conductive ceramic platelets (Supplementary Fig. 1)[32]. To enhance their interfaces with the hydrogel matrix, the ceramic platelets were modified with a silane coupling agent (3-aminopropyl)triethoxysilane (APTES)[33,34], evidenced by energy dispersive X-ray spectroscopy (EDX) mapping (Supplementary Fig. 2).

Following, 3D printable inks were prepared by mixing the pre-treated conductive ceramic platelets into a PVA solution with a small amount of Carbomer colloids as the rheology modifier[35]. Ceramic platelets and stacked Carbomer microgels formed a stiff gel structure in the composite inks, resulting in viscoelastic properties suitable for DIW 3D printing (Fig. 1b). The viscosity, storage modulus, and yield stress also increased with higher ceramic contents. Through DIW 3D printing, previous studies showed that ceramic platelets would align by the extrusion shear force in the printing nozzle, and increasing the nozzle length could improve the platelet alignment[25,29]. Figure 1c, d compared the orientation distribution of ceramic platelets in filaments extruded through a short, tapered nozzle and a 35-mm long cylindrical nozzle, respectively, as determined from their computed tomography (CT) scans (Supplementary Fig. 3). Through the long cylindrical nozzle, a significant proportion of ceramic platelets were found in the lower angle range (0–10°), which indicates their effective alignment. The scanning electron microscopy (SEM) images of a filament with 5 wt.% aligned ceramic content (CHF-5) are also presented in Fig. 1e, f, showing that the platelets aligned compliantly to the circumference of the filament. This arrangement resulted in a concentric lamellar microstructure mimicking that of the spicules of *Euplectella aspergillum*[12].

Finally, the prints were freeze-thawed twice for an initial cross-linking of the PVA matrix and further enhanced by solution substitution to obtain the final ceramic-reinforced organo-hydrogels. In the substituting solution, glycerol molecules induced strong hydrophobic aggregation of the PVA chains with abundant hydrogen bonds formed in between. Meanwhile, iron ions constructed coordination bonds by reacting with the hydroxyl (-OH) groups on the PVA chains[23]. The PVA matrix thereby gained its high cross-linking and crystallinity from the hydrogen and coordination bonds, with pure organo-hydrogel filaments (HF in Fig. 1g) already having a good tensile strength of ≈3.2 MPa, more than 30 times that of unsubstituted PVA hydrogels (≈0.1 MPa, Supplementary Fig. 4). With ceramic platelets, the composite organo-hydrogel filaments revealed significantly higher mechanical properties than those of pure organo-hydrogel (Fig. 1g, h). In particular, CHF-5 with 5 wt% ceramic platelets had the best combination of Young's modulus (≈20.3 MPa), tensile strength (≈6.9 MPa), and strain (≈347.3%). It also revealed the highest work of extension (≈17.5 MJ/m³), indicating both its excellent strength and toughness among composite hydrogels.

To investigate the effects of the above fabrication process on mechanical properties, two other groups of filaments were prepared with also 5 wt% ceramic content, but without platelet alignment (W/O Alignment), or APTES-treatment (W/O APTES). As compared in Fig. 1i, both filaments revealed significantly lower mechanical properties when compared to CHF-5, and even to pure PVA organo-hydrogel (Fig. 1g). This suggested that simply introducing untreated ceramic platelets into the organo-hydrogel compromised mechanical properties, with randomly distributed platelets and poor ceramic-hydrogel interfaces essentially creating weak points in the materials. In contrast, through an intricate combination of platelet alignment, reinforced matrix, and enhanced platelet-matrix interface, the composite design principles of natural materials were applied to the composite organo-

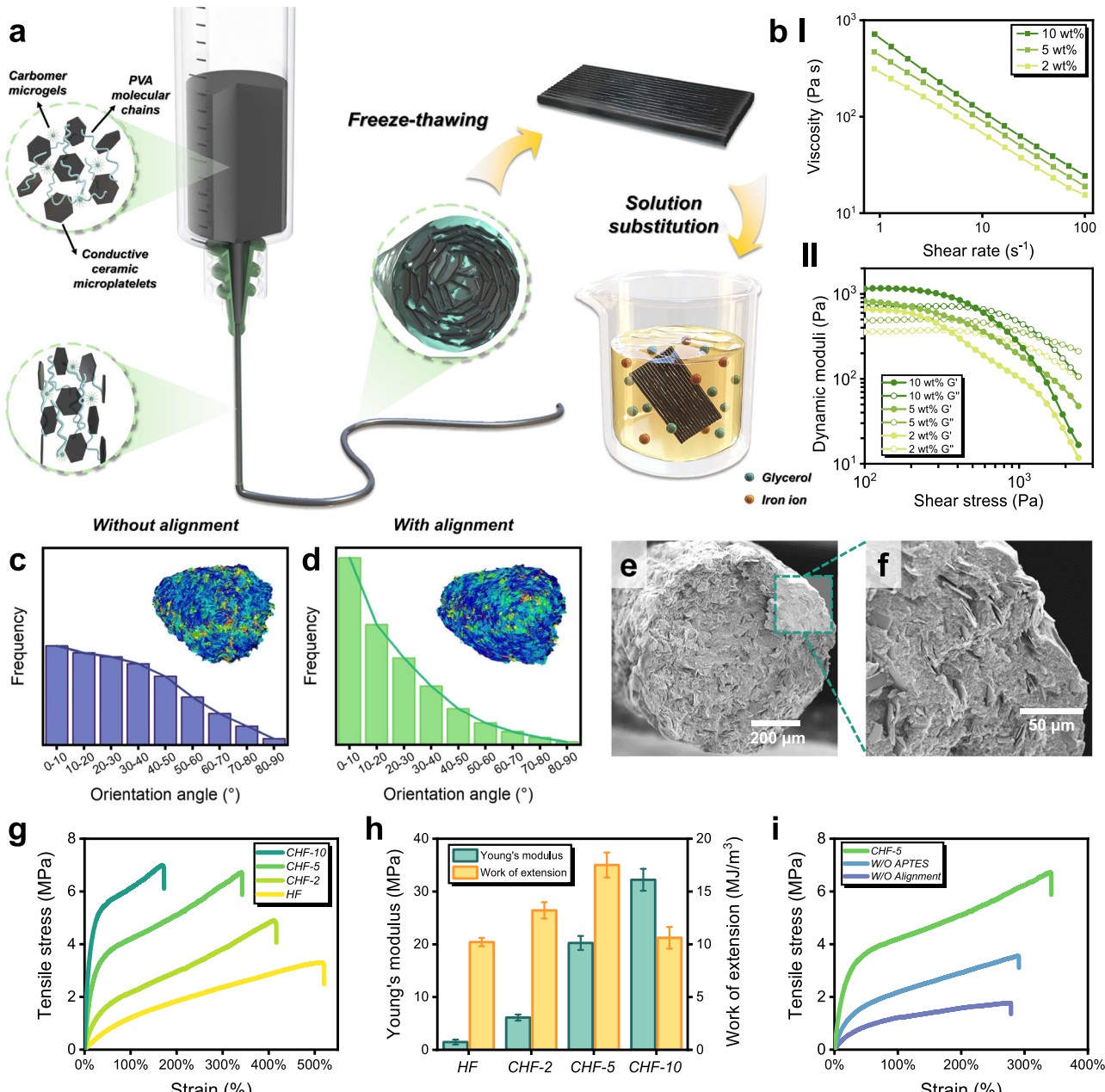

**Fig. 1 | DIW 3D printing of composite organo-hydrogel filaments. a** Schematic illustration of the fabrication process of composite organo-hydrogel filaments by shear-induced alignment in DIW 3D printing, freeze-thawing, and solution substitution. **b** The viscosity and dynamic moduli of composite inks with different ceramic contents. **c, d** Plate orientation angle analyses and CT scans of composite organo-hydrogel filaments with 5 wt.% randomly distributed and aligned platelets, respectively. **e, f** SEM images of the cross section of a composite organo-hydrogel filament with 5 wt.% aligned ceramic platelets (CHF-5). **g** Representative tensile stress-strain curves and (**h**) Young's modulus and work of extension of composite organo-hydrogel filaments with different ceramic contents. Data are presented as mean ± standard deviation from $n = 3$ independent samples. **i** Representative tensile stress-strain curves of composite organo-hydrogel filament CHF-5, with full preparation procedures, and counterparts with randomly distributed platelets (W/O Alignment) and with platelets that were not pre-treated by APTES (W/O APTES).

hydrogels, resulting in significant strengthening and toughening effects.

## DIW 3D printing of composite organo-hydrogels with bioinspired macro-architectures

Leveraging the design freedom of DIW 3D printing, the composite organo-hydrogel filaments can be easily assembled as basic building blocks to construct free-form bioinspired architectures (Supplementary Fig. 5). This process combines micro-scale reinforced filaments with macro-scale architecture design, mimicking the hierarchical organization of natural materials. To demonstrate this,

unidirectionally aligned, Bouligand, and crossed lamellar structures are selected by drawing inspiration from natural strong and tough materials (Fig. 2a–c). First, as the concentric lamellar microstructure is also present in the osteons of cortical bones, an intuitive design is to mimic their alignment by 3D printing the filaments unidirectionally. Figure 2a presents the fracture surface of the cortical bone in an elk antler, where its osteon alignment provides high stiffness and strength along the length of the antler and results in highly anisotropic mechanical properties[36]. Another example is the Bouligand architecture (Fig. 2b), a twisted plywood pattern widely found in arthropod exoskeletons with in-plane mechanical isotropy and enhanced fracture

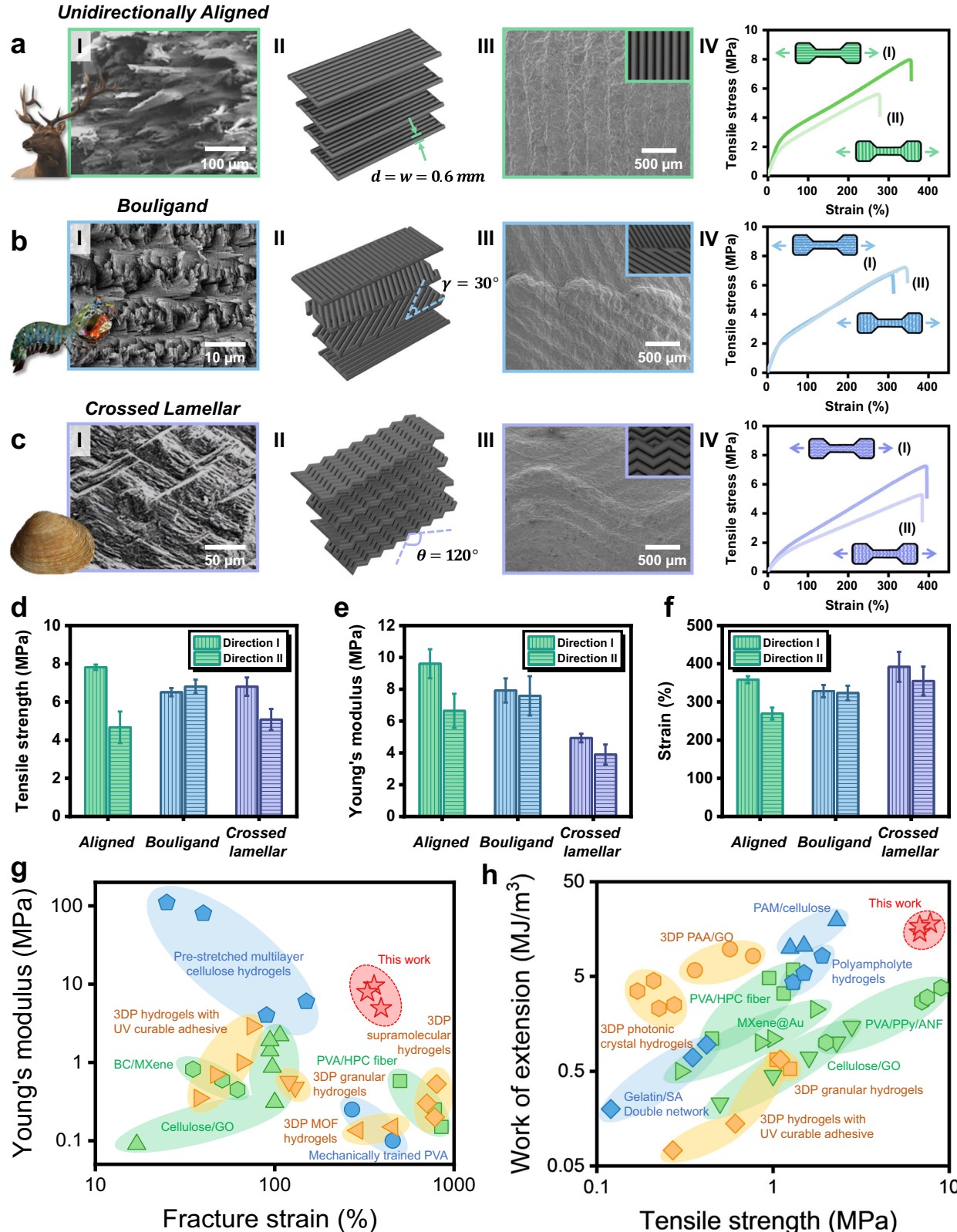

toughness[37,38]. The crossed lamellar architecture is also a bioinspired structure with excellent toughness, featuring variations in alignment orientations and an interlocking arrangement across adjacent layers (Fig. 2c)[39–41]. Based on these bioinspirations, composite organo-hydrogels were 3D printed with infill patterns as designed in Fig. 2a–c II, through the precise material deposition of DIW 3D printing (Fig. 2a–c III and Supplementary Fig. 6).

With different bioinspired macro-architectures, 3D printed composite organo-hydrogels also revealed distinct mechanical behaviors. Figure 2a–c IV compare the representative stress-strain curves of composite organo-hydrogels upon tensile loading in two normal directions. Their mechanical isotropy in tensile strength, Young's modulus, and fracture strain is compared in Fig. 2d–f. Among them, unidirectionally aligned samples (Supplementary Fig. 7a) were the

**Fig. 2 | DIW 3D printing and tensile properties of composite organo-hydrogels with different bioinspired macro-architectures. a** Unidirectionally aligned architecture mimicking the osteon alignment in the antlers of elk *C. canadensis*. Adapted from ref. 36 and reproduced with permission from Elsevier. **b** Bouligand architecture inspired by the dactyl club of mantis shrimp *O. scyllarus*. Adapted from ref. 38 and reproduced with permission from Elsevier. **c** Crossed lamellar architecture derived from the shell of bivalve mollusk *S. purpuratus*. Adapted from ref. 39 and reproduced with permission from Elsevier. Each panel includes (I) SEM image of the natural materials' microstructure, (II) model drawing of the infill pattern, (III) SEM image of the 3D printed composite organo-hydrogels, and (IV) representative stress-strain curves upon tensile loading in two normal directions. **d-f** Mechanical isotropy in tensile strength, Young's modulus, and fracture strain, respectively. Data are presented as mean ± standard deviation from $n = 3$ independent samples. **g, h** Comparison of the composite organo-hydrogels in this work with the existing literature[3,4,6,8,9,44–54] in fracture strain and Young's modulus, and tensile strength and work of extension. See Supplementary Table 1 for a complete list of data.

most anisotropic, as they exhibited the highest modulus and strength when stretched in the direction along the alignment of individual filaments, but saw significant reductions in strength, modulus, and strain in the normal direction. In the Bouligand samples, filaments were rotated at a constant $\theta = 30°$ across the layers and thus not aligned in any particular direction (Fig. 2b and Supplementary Fig. 7b). This arrangement led to an in-plane mechanical isotropy without noticeable differences in their tensile properties when loaded in two normal directions. Featuring alternating curvatures in its zig-zag filaments (Fig. 2c and Supplementary Fig. 7c), the crossed lamellar architecture was also less anisotropic, with much smaller differences in its modulus and strain between two loading directions than those of the unidirectionally aligned. These results showed that the in-plane mechanical isotropy of composite organo-hydrogels with bioinspired macro-architectures closely resembled those of their natural prototypes.

Meanwhile, three architectures also showed wide-range differences in their Young's modulus (~4–10 MPa, Fig. 2e). Unidirectionally aligned samples were most stiff when stretched along their alignment direction, while the modulus decreased in the Bouligand structure, where filaments were oriented at varying angles to the loading direction. Further, the crossed lamellar structure showed a more flexible mechanical behavior, with a much lower Young's modulus and relatively larger strain than those of the others (Fig. 2e, f), as the stretching of its zig-zag filaments also contributed to the overall deformation. Therefore, the mechanical responses, such as stiffness, of the composite organo-hydrogels can also be designed and tailored by DIW 3D printing of free-form bioinspired macro-architectures.

The combination of high stiffness, strength, stretchability, and toughness of our composite organo-hydrogels also stands out from the existing literature. As shown in Fig. 2g, previous studies on hydrogels and their composites, including both bulk fabricated and 3D printed ones, typically showed an inverse relationship between Young's modulus and fracture strain, indicating an inherent trade-off between stiffness and stretchability. In contrast, by applying the fundamental design principles of natural materials, our strategy achieved a synergistic integration of the stiff ceramic fillers and elastic hydrogel matrix, leading to both high stiffness and stretchability. The resulting composite organo-hydrogels also exhibited excellent strength and toughness (work of extension) that outperformed their counterparts in the literature, especially the 3D printed ones (Fig. 2h). 3D printed (composite) hydrogels often suffer from limited materials and processing conditions due to printability considerations, and printing defects that easily weaken the overall mechanical properties when compared to those from bulk formation processes. By incorporating both bioinspired structural and material engineering across multiple length scales, our hierarchical fabrication strategy achieves 3D printable strong and tough composite organo-hydrogels using a relatively small fraction of ceramic reinforcement (Supplementary Fig. 8). Therefore, it represents a promising approach to enhancing the mechanical properties of composite hydrogels while also introducing additional tunability in their mechanical responses.

## Enhanced fracture toughness and multi-scale mechanical energy dissipation

As the above bioinspired architectures, such as Bouligand and crossed lamellar, are also well known for their toughening effects, we further investigated the fracture toughness of composite organo-hydrogels. Pure shear tests were conducted on composite organo-hydrogels by introducing a single notch normal to the direction in which they revealed stronger tensile properties (Fig. 3a–c). Representative stress-strain curves of pre-notched PVA organo-hydrogels and composite organo-hydrogels with different macro-architectures are shown in Fig. 3d, with their fracture energy compared in Fig. 3e. Upon solution substitution, 3D printed pure PVA organo-hydrogels already had a decent fracture energy of 7.3 kJ/m², which increased about threefold (20.3 kJ/m²) in the unidirectionally aligned composite organo-hydrogels. This improvement was attributed to the stiffening and strengthening effects of aligned filaments and platelets, coupled with the crack pinning and deflection effects of the concentric lamellar microstructure[29], as suggested by the uneven fracture surface of the filaments (Supplementary Fig. 9). The Bouligand architecture saw a further increased fracture energy of 26.1 kJ/m², where the rotating filaments led to effective crack pinning and crack deflection mechanisms that delayed and deflected crack propagation (Fig. 3b, f). This also increased the energy dissipation during crack propagation, which occurred along a more tortuous crack path as observed from the fracture surface (Fig. 3b and Supplementary Fig. 10). Remarkably, the crossed lamellar samples revealed the highest fracture energy of 31.1 kJ/m². While some crack deflection was also observed (Fig. 3c), the crossed lamellar architecture led to a more prominent crack pinning effect with significantly delayed and slowed crack propagation (Fig. 3f). Due to the interlocking filament arrangement[41], it could resist crack propagation while dissipating energy by stretching its zig-zag filaments, which significantly enhanced fracture toughness. Therefore, these results demonstrate that by 3D printing bioinspired macro-architectures, our strategy can translate the toughening mechanisms of natural materials into composite organo-hydrogels to effectively enhance their fracture toughness.

By implementing the hierarchical composite design principles of natural materials, bioinspired macro-architectures were assembled from individual filaments as the fundamental building block, and each filament was intricately engineered with (1) aligned stiff platelets, (2) elastic and tough matrix, and (3) enhanced ceramic-polymer interfaces. These elements synergistically contributed to mechanical energy dissipation in both process and bridging zones across multiple length scales (Fig. 3g, h). In the process zone, the material was loaded and then unloaded as crack propagated. As illustrated in Fig. 3i, PVA matrix between aligned ceramic platelets experienced both tensile and shear loads (mostly at the overlap regions between adjacent platelets)[13]. As it facilitated load transfer between platelets via shear, the platelets also effectively carried the tensile load, resulting in higher stiffness, strength, and energy dissipation by PVA matrix deformation and friction at platelet-matrix interfaces. Simultaneously, a microscopic bridging zone formed behind the crack tip, hindering crack propagation and dissipating energy by pulling out ceramic platelets and PVA fibers (Fig. 3j). Pulled-out fibers and platelets, as well as the

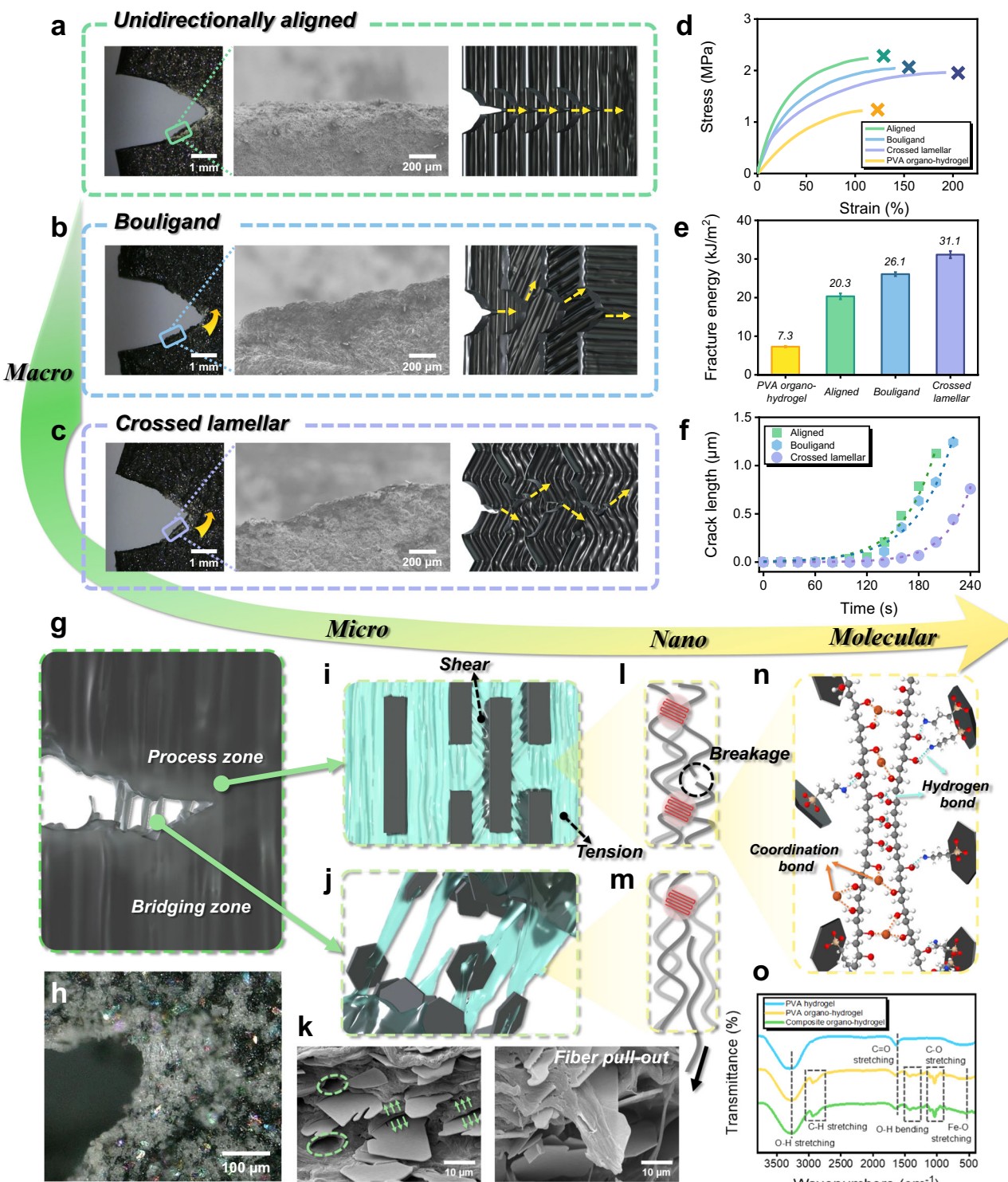

**Fig. 3 | Fracture toughness of 3D printed composite organo-hydrogels with different bioinspired macro-architectures and their multi-scale energy dissipation mechanisms.** Crack propagation in (**a**) unidirectionally aligned, (**b**) Bouligand, and (**c**) crossed lamellar samples, with each panel including an in situ optical capture upon shearing, SEM image of the crack path, and illustration of the crack propagation behavior (from left to right). **d** Representative stress-strain curves of pre-notched samples until crack propagation in pure shear tests, and (**e**) fracture energy of pure PVA organo-hydrogels and composite organo-hydrogels with different bioinspired macro-architectures. Data are presented as mean ± standard deviation from $n = 3$ independent samples. **f** Plot of crack extension in time. **g**, **h** Illustration and confocal optical image of the crack tip. **i**, **j** Illustrations of process and bridging zones at the micro scale, respectively. **k** SEM images of pulled-out ceramic platelets and PVA fibers. **l**, **m** Illustrations of PVA chain breakage and detachment from the nanocrystalline domains, respectively. **n** Illustration of hydrogen bonds (dotted lines in cyan) and iron-coordination bonds (in orange) at the molecular scale. **o** FTIR spectra of PVA hydrogel, PVA organo-hydrogel, and composite organo-hydrogel.

resulting voids, were observed at the fractured surface (Fig. 3k). Their synergistic pull-out was facilitated by enhanced platelet-matrix interfaces upon APTES-treatment, which enabled free amino ($-NH_2$) or protonated amine ($-NH_3^+$) terminal groups on platelet surfaces to form hydrogen bonding with hydroxyl groups on PVA fibers (Fig. 3n)[34].

The PVA matrix formed abundant nanocrystalline domains as high-functionality crosslinkers via solution substitution[42]. During matrix deformation and fiber pull-out, PVA chains straightened, stretched, and eventually ruptured. As multiple chains between nanocrystalline domains usually had non-uniform lengths, mechanical energy was dissipated when shorter chains were initially ruptured or detached, while longer chains still maintained the elasticity of the matrix (Fig. 3l, m)[43]. At the molecular scale, these nanocrystalline domains consisted of highly aggregated and entangled PVA chains welded by multiple molecular bonds, including hydrogen bonds between PVA chains due to hydrophobic aggregation by glycerol, and iron-oxygen coordination bonds constructed via the iron ions (Fig. 3n). Under Fourier-transform infrared (FTIR) spectroscopy (Fig. 3o), substituted organo-hydrogels revealed more distinctive peaks at ~1413 cm$^{-1}$ and ~549 cm$^{-1}$, corresponding to the bending vibration of -OH bonds and stretching vibration of iron-oxygen bonds, respectively. Higher peaks at ~2940 cm$^{-1}$ and ~1106 cm$^{-1}$ also marked stronger stretching vibrations of C-H and C-O bonds, respectively, indicating higher crystallinity of the substituted organo-hydrogels.

In summary, our strong and tough composite organo-hydrogels were achieved via a synergy of crack pinning and deflection mechanisms through 3D printed bioinspired architectures at the macro scale, coupled energy dissipation in both the process and bridging zones at the micro scale, PVA chain deformation at the nano scale, and the breakage of hydrogen and coordination bonds in the PVA matrix, as well as interfacial hydrogen bonds during platelet pull-out at the molecular scale. Instead of directly replicating the structures and compositions of a specific natural material, our strategy extracted and implemented their hierarchical composite design principles, enabling a strong coupling of inorganic and organic phases while leveraging multiple mechanisms across different length scales.

## Electrical sensing capability and wide operation tolerance

In addition to having robust mechanical properties, our hierarchical fabrication strategy also integrated multi-functionality, including electrical conductivity, strain sensing capability, and wide operation tolerance, into the composite organo-hydrogels to enhance their practical applicability. Electrical conductivity was achieved through Mg-thermic reduction of the ceramic platelets and substitution in an ionic solution. With 2 wt.% $FeCl_3$ in the substituting solution, pure PVA organo-hydrogels revealed an electrical conductivity of 5.1 S/m, which further increased to 7.1 S/m in composite organo-hydrogels at only 5 wt.% ceramic content, and up to 8.8 S/m at 10 wt.% ceramic content (Supplementary Fig. 11). These values were higher than the conductivity of conventional ion-conducting hydrogels, which mostly falls below 5 S/m[23], and the improvement was achieved by only a small addition of ceramic platelets. This suggested that the conductive ceramic platelets could effectively facilitate $Fe^{3+}$ and $Cl^-$ ion transport in the composite organo-hydrogels to enhance their electrical conductivity (Fig. 4a). Moreover, with the incorporation of these conductive ceramic platelets, 3D printed composite organo-hydrogels with different macro-architectures showed varying strain sensing properties. As compared in Fig. 4b, unidirectionally aligned samples were most sensitive to stretching with a gauge factor of ~3.75 at 200% strain, while Bouligand and crossed lamellar samples had smaller gauge factors of ~2.32 and ~1.54, respectively. Note that a similar trend was also observed in their Young's modulus, where the unidirectionally aligned architecture was the stiffest, and the crossed lamellar architecture had the lowest modulus. When composite organo-

hydrogels were stretched, their resistance changes were attributed to not only changes in the ion concentration, but also to the distancing of conductive ceramic platelets. As such, a higher stiffness implied that the ceramic platelets were more readily distanced under tension, which also resulted in larger resistance change of the composite organo-hydrogel.

The application potential of our composite organo-hydrogels was further extended by their remarkable operation tolerance, including long-term stability and wide temperature window. When exposed to the ambient environment (-25 °C) for 24 h, the composite organo-hydrogels only experienced a relative mass loss of ~12.5% (Fig. 4c). After an initial mass loss, they could reach an equilibrium state and remain stable thereafter, owing to the low evaporative pressure and hygroscopicity of glycerol in the substituting solution. The composite organo-hydrogels could also maintain stability at elevated temperatures (-60 °C), despite a higher initial mass loss of ~29%. Moreover, the thermal conductivity of organo-hydrogels was also improved with the incorporation of ceramic platelets (Fig. 4d). With a high solution content, pure organo-hydrogels had a very low thermal conductivity of 0.46 Wm$^{-1}$K$^{-1}$, which increased to 0.97 Wm$^{-1}$K$^{-1}$ at 5 wt.% ceramic content, and up to 1.23 Wm$^{-1}$K$^{-1}$ at 10 wt.% ceramic content (Supplementary Fig. 12). The enhanced thermal conductivity can help alleviate the heat buildup issue of hydrogels when they are used in flexible electronics (Supplementary Fig. 13). To the other extreme in temperature, our composite organo-hydrogels also had excellent freezing tolerance, remaining functional even at −30 °C (Fig. 4f). As revealed by the differential scanning calorimetry (DSC) results in Fig. 4g, pure PVA hydrogel had a freezing point of around −18 °C, whereas the freezing points of PVA organo-hydrogels and composite organo-hydrogels were well below −80 °C. This freezing tolerance was due to the anti-freezing properties of the glycerol-water eutectic mixture (with a mass ratio of 2:1) and high concentrations of salt (iron chloride) in the organo-hydrogels.

As shown in Fig. 4e, the composite organo-hydrogels could be used to detect various modes of deformation, such as stretching, pressing, bending, and twisting, generating stable and distinct signals of resistance change. Owing to the robust mechanical properties, they could operate under prolonged cyclic stretching and pressing (>1000 cycles, Supplementary Fig. 14) for applications that require reliable and long-lasting signal detection. Moreover, to demonstrate their versatile application in flexible electronics, composite organo-hydrogels with different bioinspired macro-architectures were integrated into a multi-functional smart sensing glove for multiple modes of sensing and detection. Figure 4h captures the configuration of the smart sensing glove, which includes conductive fingertips to allow interaction with touch screens, a pressure-sensing touch pad on the back to control robotic vehicle movement, and strain sensors on finger joints to control the gripper action. Each of these components was 3D printed with different macro-architectures to optimize their specific use: on the fingertips, composite organo-hydrogels with the in-plane isotropic Bouligand architecture were adopted to achieve uniform responsiveness across contoured fingertip surfaces; the touch pad was 3D printed with the unidirectionally aligned architecture, which was stiff and resistant to lateral deformation to ensure sensitive pressure detection; the strain sensors featured the flexible, tough, and strain sensitive crossed lamellar structure, allowing for repeated deformation detection upon bending of the finger joints. Their integration in the smart sensing glove enabled multiple modes of operation, including robotic vehicle and gripper control, and touch screen interaction (Supplementary Movie 1 and 2). This demonstration highlights the versatility of our proposed hierarchical fabrication strategy, enabling the design and 3D printing of composite organo-hydrogels with both tunable mechanical and electrical properties for specific applications.

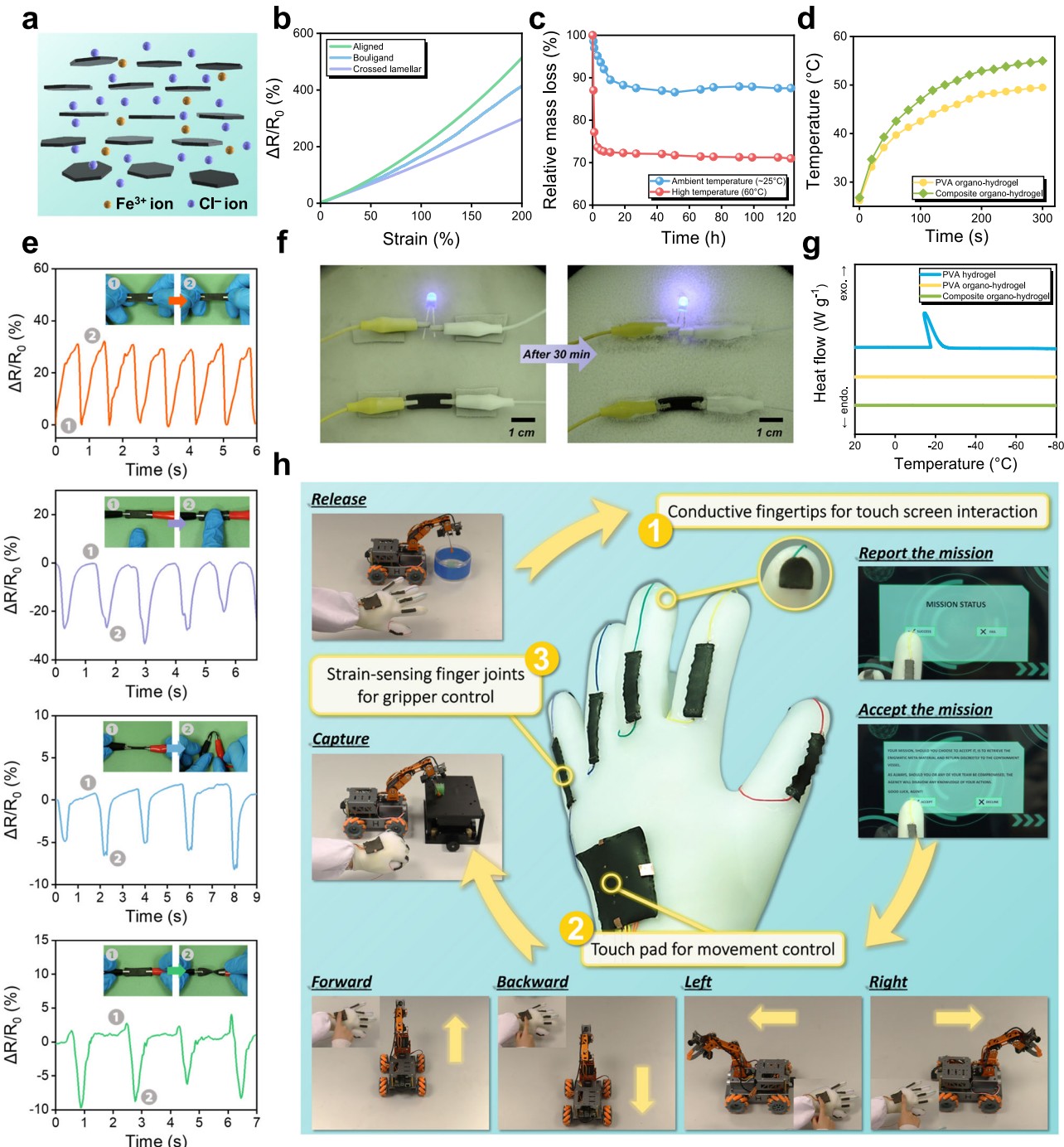

**Fig. 4 | Electrical sensing capability and operation tolerance of composite organo-hydrogels. a** Illustration of the electrical conductivity mechanism. **b** Strain sensitivity of composite organo-hydrogels with different architectures. **c** Long-term stability of the composite organo-hydrogels in terms of relative mass loss in time at ambient (−25 °C) and elevated (60 °C) temperatures. **d** Temperature increase in time of pure PVA organo-hydrogel and composite organo-hydrogel on a heating plate at 60 °C. **e** Resistance change signals of composite organo-hydrogels upon stretching, pressing, bending, and twisting (from top to bottom). **f** Lighting an LED bulb with composite organo-hydrogel on a cold surface at −30 °C. **g** Differential scanning calorimetry (DSC) results of PVA hydrogel, PVA organo-hydrogel and composite organo-hydrogel. **h** Demonstration of a multi-functional smart sensing glove that integrates conductive fingertips, touch pad, and strain sensors using the composite organo-hydrogels for touch screen interaction and robotics control.

## Discussion

By implementing the hierarchical composite design principles of natural materials, this study proposed a hierarchical fabrication design strategy for preparing strong and tough composite organo-hydrogels intricately engineered with (1) 5 wt.% aligned conductive ceramic platelets, (2) a highly crystalline PVA matrix, and (3) silane-treated platelet-matrix interfaces. The composite organo-hydrogels were also 3D printed with a selection of bioinspired macro-architectures, leading to a combination of high stiffness (up to 9.6 MPa), strength (up to 7.8 MPa), fracture energy (up to 31.1 kJ/m²). In particular, the excellent fracture toughness was achieved through synergistic energy dissipation in both process and bridging zones across multiple length scales. With enhanced electrical (7.1 S/m) and thermal conductivity (0.97 W m⁻¹K⁻¹), as well as wide operation tolerance, these composite

organo-hydrogels could be used for various mechanically demanding applications, including flexible electronics. Hence, the proposed strategy effectively leveraged the hierarchical composite design principles of natural materials, achieving bioinspired mechanical mechanisms, and tunable mechanical and electrical sensing responses in the composite organo-hydrogels. The groundwork laid by this model strategy opens exciting opportunities for the development of advanced composite hydrogels. The versatility of DIW 3D printing in both material and structural design can be further exploited. Therein lies the potential to develop composite hydrogels with novel bioinspired architectures, further exploring the hierarchical organization of natural materials for enhanced performance.

## Methods

### Pre-treatment of ceramic platelets
Conductive ceramic platelets were prepared via a magnesium reduction process. 3 g of the titania-coated alumina platelets (Xirallic® T50-10) were thoroughly mixed with 1 g of magnesium (Mg) powder before heat treatment at 650 °C for 3 h under an argon atmosphere. The platelets were then washed with 0.1 M HCl solution to remove Mg residual and cleaned with deionized (DI) water three times before collected by centrifugation. For surface modification with (3-aminopropyl)triethoxysilane (APTES), 10 g of the platelets were added to 100 mL of DI water with 10 mL of APTES, and the mixture was stirred at room temperature for 24 h. The APTES-modified conductive ceramic platelets were washed three times with DI water, centrifuged, and dried before use.

### DIW 3D printing of ceramic-reinforced composite organo-hydrogels
Poly(vinyl alcohol) (PVA, $M_w$ = 146 ~ 186 kDa) hydrogel ink was prepared by adding 2 wt.% Carbomer powder into an 8 wt.% PVA solution to achieve suitable rheological behaviors. An adequate amount of the pre-treated ceramic platelets (2, 5, and 10 wt.%) was then added to formulate the composite hydrogel ink. Each addition to the mixture was followed by homogenization at 2000 rpm for 5 min in a planetary mixer (Kakuhunter SK-300SII). The Allevi 2 bioprinter was used to 3D print the composite hydrogel ink using a cylindrical nozzle (inner diameter: 0.62 mm, length: 35 mm) at a constant printing speed of 4 mm/s under pneumatic pressure. The printed composite hydrogel samples were freeze-thawed twice and subsequently immersed in a glycerol-water-iron chloride mixture for 24 h at room temperature. The solution was prepared by dissolving 2 wt.% iron chloride powder in a glycerol-water mixture at a 2:1 weight ratio. After solution substitution, composite organo-hydrogel samples were obtained.

### Rheology measurement
A shear rate and stress-controlled rheometer (HAAKE MARS, Thermo Fisher Scientific) was used for rheology characterization. The composite hydrogel ink was filled between a pair of 35- mm parallel plates at a 0.10- mm gap. The viscosity was measured against increasing shear rates from 0.1 to 100 s$^{-1}$, and dynamic moduli were measured by oscillatory tests at 1 Hz frequency with increasing stresses from 100 to $10^4$ Pa.

### Material characterizations
The elemental composition of pre-treated ceramic platelets was studied via energy-dispersive X-ray (EDX) analysis and X-ray photoelectron spectroscopy (XPS). The microstructure of the platelets and 3D printed composite organo-hydrogels were observed using a field emission scanning electron microscope (FE-SEM S-4300, Hitachi). X-ray computed tomography (CT) scans were performed using ZEISS Xradia 520 Versa to characterize the platelet orientation and distribution in the composite organo-hydrogels. The molecular bonds in the composite organo-hydrogels were studied by Fourier transform infrared spectroscopy with an attenuated total reflectance module (ATR-FTIR, Agilent CARY 660). To study the freezing tolerance of the composite organo-hydrogels, differential scanning calorimetry (DSC) was performed in air from room temperature to −90 °C at a constant rate of −5 °C/min to identify their freezing points.

### Mechanical testing
Tensile and pure shear tests were performed with an Instron 5500 Micro Tester at a constant loading rate of 10 mm/min. Tensile test specimens were 3D printed into a standard dumb-bell shape with a final gauge length of ~10 mm, width of ~5 mm, and thickness of ~1.2 mm. The work of extension was calculated by integrating the area under the tensile stress-strain curves until the strain-to-failure $\varepsilon_f$ via the equation below:

$$W = \int_0^{\varepsilon_f} \sigma \cdot d\varepsilon \tag{1}$$

For pure shear tests, the specimens were 3D printed with a final gauge length of ~7.5 mm, width of ~15 mm, and thickness of ~1.2 mm. An initial cut of 5 mm was introduced at the middle point of the gauge length to prepare singled-notched specimens. By stretching the notched specimens, the critical strain at the occurrence of unstable crack propagation ($\varepsilon_c$) was obtained from their strain at maximum stress. Then, the fracture energy was calculated via the equation below by multiplying the gauge length (H) with the area under the stress-strain curve of the unnotched specimen until the critical strain $\varepsilon_c$.

$$\Gamma = H \int_0^{\varepsilon_c} \sigma \cdot d\varepsilon \tag{2}$$

### Thermal conductivity characterization
A thermal conductivity analyzer (Trident, CTherm) with a modified transient plane source (MTPS) was used to measure the thermal conductivity of the composite organo-hydrogels that were 3D printed with final dimensions of ~ $20 \times 20 \times 3$ mm³. To study their thermal conduction behavior, the specimens were placed on a heating platform at 60 °C, with the temperature increases monitored by a handheld infrared camera (M600, InfiRay) to capture infrared images in real time.

### Electrical conductivity and strain sensing characterization
The electrical conductivity of the composite organo-hydrogels was obtained using four-point conductivity probe measurement. To demonstrate the conductivity of the organo-hydrogels, light-emitting diode (LED) light bulbs and connecting wires with crocodile clips were used to build the circuit set-up with a 3 V voltage source. For sensing applications, the real-time resistance change of the composite organo-hydrogels under different deformation modes was recorded by a digital multimeter (Keithley DMM6500). The smart sensing glove was made by stitching the composite organo-hydrogels on a silicone glove. The resistance change signals were collected by a development board (NodeMCU ESP-32S) and identified using a self-built Python code to send programmable commands to a robotic vehicle. Informed consent was provided by the research participant in Fig. 4h.

### Statistics and reproducibility
Representative results in Figs. 1e, f, 2a–c, and 3, and in Supplementary Figs. 7, 9, and 10 were obtained after three independent experiments showing similar results.

### Reporting summary
Further information on research design is available in the Nature Portfolio Reporting Summary linked to this article.

## Data availability

The data that support the findings of this study are available from the corresponding author upon request.

## Code availability

The custom code for detecting resistance changes and controlling the robotic vehicle is available from the corresponding author upon request.

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

## Acknowledgements
This work was supported by the Singapore Ministry of Education Academic Research Fund Tier 2 Grant (grant number MOE-T2EP50122-0007). The authors thank Ms. Yuchun Ji from Guilin University of Technology for providing valuable computed tomography scans of our materials.

## Author contributions
Q.L., X.D., and W.Z. conceived the research. Q.L. and T.L. pre-treated ceramic platelets. T.L. and G.L. characterized and analyzed the pre-treated ceramic platelets. Q.L. and X.D. designed and prepared the composite organo-hydrogels. Q.L. and X.D. performed structural and compositional characterizations. H.Z. performed X-ray CT scans of the materials. Y.Z. measured the electrical conductivity of the materials. Q.L., X.D., and H.Q. designed and performed the electrical sensing demonstration of the materials. Q.L. wrote the original draft. X.D. and W.Z. revised the manuscript. W.Z. supervised the research.

## Competing interests
The authors declare no competing interests.
