## [Peer Review File · Nature Communications]

nature portfolio

Peer Review File

3D printable strong and tough composite organo-hydrogels inspired by natural hierarchical composite design principlesReviewers' comments:

Reviewer #1 (Remarks to the Author):

Zhai and coworkers reported the fabrication of bioinspired architectures from the integration of ingredients of direct-ink-write (DIW) 3D printing, shear-induced alignment, solvent-induced gelation of poly(vinyl alcohol) (PVA) matrix, etc. They claimed that the as-fabricated composite organo-hydrogels demonstrated the combination of high stiffness, strength, toughness, high electrical conductivity, as well as wide-range operation tolerance. However, this work does not carry the novelty that Nature Communications is looking for, since

1. Functional materials/architectures from DIW 3D printing by exploiting the shear-induced alignment has been widely reported, such as *Advanced Functional Materials* 27.12 (2017): 1604619; *Proceedings of the National Academy of Sciences* 115.6 (2018): 1198-1203; *Composites Part B: Engineering* 229 (2022): 109479; *Advanced materials* 30.10 (2018): 1706164; *Nature communications* 6.1 (2015): 8643; *Nature* 561.7722 (2018): 226-230, to cite only a few.
2. They claim that they could fabricate the unidirectionally aligned, Bouligand, and crossed lamellar structures, as shown in Figure 2. However, as shown in Figure 3, the mechanics did not follow the structure-property relation as demonstrated in those natural prototypes, this is because the length scales are not the same level in Figure 2. Therefore, such kind of bioinspired fabrication is too superficial, and could not deliver enough information for new material design and fabrication.
3. The author claimed the synergistic contribution of hierarchical structure to the high toughness. However, how to justify the synergy has not been clearly disclosed. There are few length scales within the systems. Nano: polymer chains, nanocrystal, nanoparticles, and nanoparticle-polymer chain interaction; Micro: phase separation, microscale alignment; Macro: 3d printed structures, geometries, filament-filament interaction. They should take all these into consideration.
4. They claim the term of organo-hydrogels, what is the exactly phase within these composite materials, without these information, one can not exactly specify the structure-properties relationship.
5. Demonstration in Figure 5 is not convincing. Why such composite organo-hydrogels are necessary for such kind of sensing gloves, is the combination of high stiffness, strength, toughness, high electrical conductivity, as well as wide-range operation tolerance, indispensable for such kind of sensing gloves? The previous 4 figures are dealing with mechanics, why it changed to electrical performance? And how 3DP play its role in such an application scenario? Without these rationales, Figure 5 is just a stacking of data and images.

Reviewer #2 (Remarks to the Author):

The manuscript reports on the 3D printing of an organo-hydrogel reinforced with 5 wt% ceramic microplatelets. The microplatelets have had their surface modified to enable strong binding with the hydrogel. Direct ink writing is the 3D printing method used. Due to the shear developing during the

printing, circular alignment of the microplatelets is found. The authors use this method to reproduce natural microstructures. They measure the mechanical, electrical, and thermal properties of the resulting composites and suggest their use for flexible electronics.

The manuscript is well structured and interesting. However, there are a number of claims that need to be toned down. There are few inaccuracies in the introduction. Also, the novelty of the work is questionable, mostly related to the use of the organo-hydrogel as a matrix. The authors claim several times the novelty of their work, but there are other similar studies published. Also, the authors claim their materials to be bioinspired with natural microstructures but the concentration of mineral elements is 5 wt%, as compared to more than 60 vol% in natural composites.... Finally, the authors characterized the properties and mention applications, however, they overestimate the potential of their materials and overclaim their potential applications. Some more details can also be given in the results section. Because of these overclaims and because the literature in the field is not properly covered, I cannot recommend this manuscript for Nature Communications journal.

I detail my comments below:

- Introduction, page 2, line 24: 'intrinsically fragile structures' -> This is inaccurate, Interpenetrated hydrogels are not fragile structure.
- Page 2, line 38: for 'magnetic' could also refer to this work on reinforced hydrogels: DOI 10.1088/1748-3190/acd42d
- Page 2, line 39. What does 'mechanical training' means?
- Page 2, line 41: 'their high energy consumption'. Disagree with the high energy consumption: it is compared to what? Self-assembly methods are rather low energy...
- Page 2, line 43: 'lack of elaborate, localized microstructural control to replicate the complex hierarchical microstructures observed in natural materials'. I think this is a bold and unpleasant statement. First this work does not particularly reproduce the natural microstructures found in natural materials. Second, there are methods that can do this quite well like freeze casting with controlled temperature gradient (DOI: 10.1126/sciadv.1500849), magnetically assisted slip casting (<https://doi.org/10.1007/s11837-023-05705-w>), or even 3d printing using a large concentration of microplatelets (DOI<https://doi.org/10.1038/s41598-017-14236-9>).
- Page 3, line 60: "novel fabrication strategy that achieves multi-scale coupling effects by integrating material processing techniques on the molecular and nano scale, and DIW 3D printing of hierarchical bioinspired architectures on the micro and macro scales" ->The claim of novelty here is inaccurate. There are other works published that do that already, for example: DOI<https://doi.org/10.1038/s41586-018-0474-7>, DOI<https://doi.org/10.1038/nmat4544>
- Page 3, line 63: the authors use ceramic microplatelets but in the introduction, they only mention the use of DIW to align microfibers along the printing direction. Because microplatelets are 2D and because some methods have been developed to create complex microstructures using DIW and controlled microplatelets orientations, these 3D printing methods should also be reviewed.
- Page 3, line 65: Concentric lamellar microstructures are also not new and may not enable the control over multiple architectures. Other papers that report on this microstructure: DOI<https://doi.org/10.1038/s41598-017-14236-9>, DOI: 10.18063/ijb.v8i2.551
- Page 4, line 69: 'freeze-thawing twice": since the authors are also using freezing, how is that less energy intensive as the other methods they were reporting on earlier?

- Page 4, line 74: For the application for flexible electronics, do the final material contain water?
- Page 6, line 108: the ceramic content is 5 wt%. It would be helpful to talk in terms of vol% to be able to compare with natural materials, which typically have between 60 and 96 vol% of mineral content, including the spicules the authors are referring to.
- Page 7, line 124: ‘excellent strength and toughness’’: in comparison to what? It looks pretty low in comparison to metals, for example..
- Page 7, line 135: “interlocked”. Given the extremely low concentration in ceramic microplatelets, is there any interlocking?
- Page 8, line 147: “cortical bones”. Cortical bones have more than 60 vol% mineral hence cannot compare with the current material.
- Figure 2, c, III: is the schematic adequate? Usually, 3D printers cannot print with control over the Z-direction. It does not correspond to the schematic in Figure 4, f.
- Figure 3: the property maps should include hydrogels with microplatelets and other microplatelet-based organic composites.
- Page 18, line 315: How much water is there in the final composites? What was the humidity of the environment in which the relative mass loss tests were conducted?
- Page 18, line 321: 0.97 W/mK for thermal conductivity is extremely low, how can it be suggested that it could help the thermal dissipation in flexible electronics? This is strong overclaim. The electrical conductivity of 7.1 S/m is also very low.
- Page 21, line 362: “novel class” -> please remove the claim of novelty.

Reviewer #3 (Remarks to the Author):

Comments

This paper reported a direct ink writing method to replicate the biological hierarchical structure. By extruding the composite hydrogel ink, the ceramic micro-platelets were aligned along the extrusion direction, and the concentric-circle-oriented structure was obtained. The organo-hydrogel was used to print composites with three bioinspired structural elements. The mechanical properties, such as stiffness, strength, and fracture energy, of the organo-hydrogels were enhanced by the multiscale energy dissipation mechanism. The composite prepared by the proposed method was extended to flexible electronics and validated for applications in robot-vehicle control and touch-screen interaction. However, major concerns listed below are needed to be clarified before this work is publishable.

1. The work is conceptually similar to Ref. 30 (Sci. Rep. 7, 13759 (2017)) or Adv. Mater. 35, 2211175 (2023). In these references, they also extruded composite inks containing ceramic platelets to obtain concentric circle-oriented lines and print the bioinspired Bouligand structure. They got the similar finding that bioinspired architectures increase energy absorption during fracture. Additional experiments and notes should be included to clarify what new conclusions have been developed compared to previous studies.

2. In direct ink writing technique, the shear field induced by extrusion does not necessarily imply complete orientation of the platelets. Optimizing the printing parameters (flow rate, nozzle diameter, etc.) and ink composition ratios, as well as detailed characterization of the orientation degree of the platelets, would enhance the credibility of the article.
3. Fig. 2a(IV), b(IV), and c(IV) displayed SEM images where the layered interface might be a result of the printed lines, and the arrangement and distribution of ceramic platelets are still ambiguous. Finer 3D structural characterization of the printed hierarchical architectures will make the structure clearer.
4. Fig.3d and Fig. 3e should include more mechanical gel composites, such as anisotropic PVA hydrogel (Nature 590, 594-599 (2021)), alginate hydrogels (Nat. Commun. 13, 3019 (2022)), PVA/aramid nanofiber hydrogel (Nat. Commun. 14, 759 (2023)), etc.
5. The authors emphasize that direct ink writing can produce hierarchical composites that exhibit superior mechanical properties. However, in terms of electronic devices, it is necessary to compare the performances with those of other architected electronic devices to highlight the advantages of the biomimetic hierarchical structures.
6. On Page 8, Line 157-158, "... demonstrated the precise material deposition of DIW 3D printing and excellent shape fidelity of the composite hydrogel ink." When discussing shape fidelity, quantitative data is needed to compare the discrepancies between the designed structure and the printed structure.

Dear reviewers,

We deeply appreciate your time and effort in reviewing our initial manuscript and providing constructive feedback. After careful consideration of reviewers' comments, we acknowledge that our original manuscript may not have adequately presented the novelty and significance of our research. In response, we have thoroughly revised our manuscript to present the research significance that distinguish our work from existing literature more effectively. We are grateful for the invaluable insights provided by the reviewers, which have been instrumental in our comprehensive revision process. These improvements have significantly enhanced the potential of our work for publication. We are hopeful for the opportunity to have our manuscript re-evaluated, confident that this revised submission, now enriched with a fresh perspective and additional results and analysis, will align with the high standards of *Nature Communications*.

Reviewer #1 (Remarks to the Author):

Zhai and coworkers reported the fabrication of bioinspired architectures from the integration of ingredients of direct-ink-write (DIW) 3D printing, shear-induced alignment, solvent-induced gelation of poly(vinyl alcohol) (PVA) matrix, etc. They claimed that the as-fabricated composite organo-hydrogels demonstrated the combination of high stiffness, strength, toughness, high electrical conductivity, as well as wide-range operation tolerance. However, this work does not carry the novelty that Nature Communications is looking for, since

1. Functional materials/architectures from DIW 3D printing by exploiting the shear-induced alignment has been widely reported, such as *Advanced Functional Materials* 27.12 (2017): 1604619; *Proceedings of the National Academy of Sciences* 115.6 (2018): 1198-1203; *Composites Part B: Engineering* 229 (2022): 109479; *Advanced materials* 30.10 (2018): 1706164; *Nature communications* 6.1 (2015): 8643; *Nature* 561.7722 (2018): 226-230, to cite only a few.

Thank you for pointing out the extensive research on shear-induced alignment by DIW 3D printing. Our work indeed builds upon the well-explored technique of shear-induced assembly by DIW 3D printing, which, as rightly noted, has been a focus of several high-

impact studies. However, we wish to emphasize that the novelty of our study lies in its bioinspired multi-scale fabrication strategy, which not only leverages (1) filler alignment by DIW 3D printing but also encompasses (2) hydrogel matrix reinforcement by solution substitution and (3) ceramic-polymer interface treatment. This strategy is fundamentally inspired by the hierarchical composite design principles observed in natural composite materials, particularly in hard biological tissues. These natural composites showcase a synergistic integration of (1) aligned stiff anisotropic particles or fibers, (2) embedded in a soft and tough matrix, (3) with a tight interface between the stiff elements and the soft matrix is observed. Accordingly, our fabrication strategy implements these bioinspired designs in our composite organo-hydrogel filaments, which can be utilized as fundamental building blocks for 3D printing various bioinspired hierarchical structures with their representative mechanical behaviors and mechanisms. Therefore, our bioinspired bottom-up approach is versatile and represents a novel concept not yet explored or developed in the cited literature.

2. They claim that they could fabricate the unidirectionally aligned, Bouligand, and crossed lamellar structures, as shown in Figure 2. However, as shown in Figure 3, the mechanics did not follow the structure-property relation as demonstrated in those natural prototypes, this is because the length scales are not the same level in Figure 2. Therefore, such kind of bioinspired fabrication is too superficial, and could not deliver enough information for new material design and fabrication.

We appreciate the reviewer's perspective that the structure-property relation of natural prototypes should be achieved by replicating their microstructures to the exact length scales in our materials. While this is certainly an exciting and promising avenue of research, our bioinspired fabrication strategy intentionally diverges from it. Instead, this work aims to abstract and implement the fundamental hierarchical composite design of natural materials. Although such design principles are predominantly observed in hard biological tissues, our findings demonstrate their applicability and effectiveness in reinforcing soft materials such as our composite organo-hydrogels. In Fig. 2 and 3 of the revised manuscript, we have shown how different bioinspired structures can affect the material strength, anisotropy, and fracture toughness. Although we acknowledge that our composite organo-hydrogels do not

exactly replicate the mechanical characteristics of their hard biological counterparts, our work effectively achieves similar bioinspired mechanical mechanisms to enhance material strength and toughness. As such, the term “superficial” can be viewed as an advantage, indicating that bioinspired mechanical mechanisms can be achieved through practical and accessible engineering methods. As now discussed in the revised manuscript, this approach highlights our contribution to bioinspired material engineering as a distinct methodological perspective, and the mechanical insights contribute to a broader application of bioinspired design principles in material engineering.

3. The author claimed the synergistic contribution of hieratical structure to the high toughness. However, how to justify the synergy has not been clearly disclosed. There are few length scales within the systems. Nano: polymer chains, nanocrystal, nanoparticles, and nanoparticle-polymer chain interaction; Micro: phase separation, microscale alignment; Macro: 3d printed structures, geometries, filament-filament interaction. They should take all these into consideration.

Thank you for your suggestion to provide a more detailed discussion on the synergistic contributions of hierarchical structures to the toughness of our materials. In our revised manuscript, we have elucidated the synergistic effects from two perspectives. Firstly, the high toughness of our composite organo-hydrogels is attributed to applying the hierarchical composite design principles – the synergy of (1) filler alignment, (2) matrix reinforcement, and (3) interfacial treatment. Secondly, we have also discussed the multi-scale mechanisms across macro, micro, nano, and molecular scales. Fig. 3 is also refined with more detailed illustrations to support our discussions. For your convenience, below are key excerpts from the revised manuscript that summarize our analyses on the synergistic effects. We also kindly invite you to review the detailed discussions in our revised manuscript and would greatly appreciate any additional suggestions or insights that can help us improve.

By implementing the hierarchical composite design principles of natural materials, bioinspired macro-architectures were assembled from individual filaments as the fundamental building block, and each filament was intricately engineered with (1) aligned stiff platelets, (2) elastic and tough matrix, and (3) enhanced ceramic-polymer interfaces.

These elements synergistically contributed to mechanical energy dissipation in both process and bridging zones across multiple length scales (Figure 3g, h).

In summary, our strong and tough composite organo-hydrogels were achieved via a synergy of crack pinning and deflection mechanisms through 3D printed bioinspired architectures at the macro scale, coupled energy dissipation in both the process and bridging zones at the micro scale, PVA chain deformation at the nano scale, and the breakage of hydrogen and coordination bonds in the PVA matrix, as well as interfacial hydrogen bonds during platelet pull-out at the molecular scale. Instead of directly replicating the structures and compositions of a specific natural material, our strategy extracted and implemented their hierarchical composite design principles, enabling a strong coupling of inorganic and organic phases while leveraging multiple mechanisms across different length scales.

Fig. 3. g-h) Illustration and confocal optical image of crack tip. **i-j)** Illustrations of process and bridging zones at the micro scale, respectively. **k)** SEM images of pulled-out ceramic platelets and PVA fibers. **l-m)** Illustrations of PVA chain breakage and detachment from the nanocrystalline domains, respectively. **n)** Illustration of hydrogen bonds (dotted lines in cyan) and iron-coordination bonds (in orange) at the molecular scale. **o)** FTIR spectra of PVA hydrogel, PVA organo-hydrogel, and composite organo-hydrogel.

4. They claim the term of organo-hydrogels, what is the exactly phase within these composite materials, without these information, one can not exactly specify the structure-properties relationship.

Thank you for your inquiry about the compositions of our materials. As detailed in the manuscript, our composite PVA organo-hydrogels consist of 5 wt.% ceramic platelets and are substituted by a water-glycerol (1:2) binary solvent with 2 wt.% ferric chloride. FTIR analysis is also provided to characterize their molecular composition.

5. Demonstration in Figure 5 is not convincible. Why such composite organo-hydrogels are necessary for such kind of sensing gloves, is the combination of high stiffness, strength, toughness, high electrical conductivity, as well as wide-range operation tolerance, indispensable for such kind of sensing gloves? The previous 4 figures are dealing with mechanics, why it changed to electrical performance? And how 3DP play its role in such an application scenario? Without these rationales, Figure 5 is just a stacking of data and images.

We appreciate the opportunity to clarify the intention and significance behind Fig. 5 (now Fig. 4 in the revised manuscript) and its relevance to the objectives of our study. While this work primarily aims to develop strong and tough composite organo-hydrogels via bioinspired hierarchical composite designs, flexible electronics has been the targeted application from the start. Conductive materials with excellent mechanical properties such as high strength and high toughness are highly sought-after in flexible electronics. This is why our fabrication process incorporates ceramic platelet treatment via Mg-reduction and the addition of conductive ferric chloride ions to the water-glycerol binary solvent. Their incorporation also showcases the versatility of our process in enhancing the functionality of both ceramic and PVA hydrogel components. Therefore, the transition in our manuscript from discussing mechanical to electrical properties underscores the multifunctionality of our composite organo-hydrogels.

Followingly, the sensing glove demonstration prepares composite organo-hydrogels with varying 3D printed architectures, and thus varying mechanical and sensing properties, and strategically combines them at different locations to fulfill distinct sensing functions. This design rationale is detailed in the manuscript, which showcases the capability of our

3D printing-based strategy in achieving tailored material properties and functionalities for specific application scenarios, such as flexible electronics. Therefore, this section serves as an important demonstration of our material's critical attributes – high stiffness, strength, toughness, and electrical conductivity with a wide operational tolerance, illustrating the practical relevance and potential of our research.

Reviewer #2 (Remarks to the Author):

The manuscript reports on the 3D printing of an organo-hydrogel reinforced with 5 wt% ceramic microplatelets. The microplatelets have had their surface modified to enable strong binding with the hydrogel. Direct ink writing is the 3D printing method used. Due to the shear developing during the printing, circular alignment of the microplatelets is found. The authors use this method to reproduce natural microstructures. They measure the mechanical, electrical, and thermal properties of the resulting composites and suggest their use for flexible electronics.

The manuscript is well structured and interesting. However, there are a number of claims that need to be toned down. There are few inaccuracies in the introduction. Also, the novelty of the work is questionable, mostly related to the use of the organo-hydrogel as a matrix. The authors claim several times the novelty of their work, but there are other similar studies published. Also, the authors claim their materials to be bioinspired with natural microstructures but the concentration of mineral elements is 5 wt%, as compared to more than 60 vol% in natural composites. Finally, the authors characterized the properties and mention applications, however, they overestimate the potential of their materials and overclaim their potential applications. Some more details can also be given in the results section. Because of these overclaims and because the literature in the field is not properly covered, I cannot recommend this manuscript for Nature Communications journal.

Dear reviewer,

Thank you for your interest in this work and constructive feedback on our manuscript. We have thoroughly reviewed your comments and made significant revisions to address

your concerns. Below is a summary of our responses to the major concerns raised, with a detailed point-by-point response also provided for your review.

To highlight the novelty of our work, we have completely revised both the abstract and introduction. These sections now clearly explain the unique design rationale of our approach, differentiating our work from existing studies, especially those on shear-induced alignment by DIW 3D printing. Specifically, our work proposes a comprehensive multi-scale fabrication strategy that not only leverages (1) filler alignment by DIW 3D printing but also encompasses (2) solution substitution for hydrogel matrix reinforcement and (3) ceramic-polymer interface treatment. As discussed in the revised manuscript, our strategy is fundamentally inspired by the hierarchical composite design principles observed in natural materials, which integrate (1) aligned stiff anisotropic particles or fibers, (2) in a soft and tough matrix, (3) with a tight interface between the stiff elements and the soft matrix. Accordingly, our fabrication strategy implements these bioinspired designs in our composite organo-hydrogel filaments, which can be then utilized as fundamental building blocks for 3D printing various bioinspired hierarchical structures with their representative mechanical behaviors and mechanisms. Therefore, our bioinspired approach represents a novel concept not yet explored or developed in the cited literature.

These clarifications also answer your question about our bioinspired approach, such as the drastic difference between the ceramic content of our composite organo-hydrogels and natural materials. Indeed, instead of exactly replicating the composition and structure of natural materials, this work aims to abstract and implement their hierarchical composite design. While these principles are commonly found in hard biological tissues with a large ceramic content, our results demonstrate their applicability and effectiveness in enhancing the strength and toughness of soft materials such as composite hydrogels. Therefore, we believe this also highlights our distinct methodological perspective to bioinspired material engineering, and the mechanical insights contribute to a broader application of bioinspired design principles in material engineering.

In the meantime, we have corrected any inaccuracies and included relevant references in the introduction as recommended. We have also critically reviewed our claims regarding the application potential of our materials. Any overestimations and overclaims have been

rectified to reflect the material's capabilities and potential uses in flexible electronics more accurately. We believe these revisions have significantly strengthened our manuscript and hope that it now meets the standards of *Nature Communications*. We are grateful for the opportunity to improve our work based on your invaluable feedback and look forward to the possibility of our revised manuscript being re-considered for publication.

I detail my comments below:

- Introduction, page 2, line 24: 'intrinsically fragile structures' -> This is inaccurate, Interpenetrated hydrogels are not fragile structure.

Thank you for pointing out this issue. Our intent was to describe that conventional hydrogels have fragile structures, while we assumed an understanding that interpenetrating double network hydrogels serve as a mechanical reinforcement method. To clarify, we have revised the sentence as follows:

Hydrogels are three-dimensional polymeric networks that can retain large amounts of water, but conventional hydrogels with limited cross-linking and loose polymer networks are relatively weak and fragile to meet the demands of real-life applications¹.

1. Hua, M. et al. Strong tough hydrogels via the synergy of freeze-casting and salting out. *Nature* 590, 594–599 (2021).

- Page 2, line 38: for 'magnetic' could also refer to this wok on reinforced hydrogels: DOI 10.1088/1748-3190/acd42d

Thank you for your comment. We have included this reference in the introduction as below.

To this end, various structural engineering approaches have been developed to fabricate composite hydrogels with anisotropically aligned microstructures¹⁶, such as using magnetic^{5,17,18} and electric fields^{7,19}, mechanical training^{20,21}, freeze casting^{22,23}, and self-assembly²⁴.

18. Sapasakulvanit, S., Chan, X. Y. & Le Ferrand, H. Fabrication and testing of bioinspired microstructured alumina composites with sacrificial interpenetrating polymer bonds. *Bioinspir. Biomim.* 18, 046009 (2023).

- Page 2, line 39. What does ‘mechanical training’ mean?

Mechanical training is a process in which hydrogels are subjected to mechanical forces or deformation, such as stretching, which can align the polymer fibers in the loading direction and enhance their mechanical properties.

- Page 2, line 41: ‘their high energy consumption’. Disagree with the high energy consumption: it is compared to what? Self-assembly methods are rather low energy...

Thank you for your comment. By “high energy consumption” we were referring to the freeze casting process and the use of external fields, since freeze casting often utilizes liquid nitrogen or large cooling systems, while creating external fields requires specialized equipment. Indeed, this claim overlooked the relatively low energy demands of self-assembly and the energy intensity of our own 3D printing method. We acknowledge that this statement was imprecise and not essential to our argument, and we have omitted it to ensure accuracy and relevance in our discussion.

- Page 2, line 43: ‘lack of elaborate, localized microstructural control to replicate the complex hierarchical microstructures observed in natural materials. I think this is a bold and unpleasant statement. First this work does not particularly reproduce the natural microstructures found in natural materials. Second, there are methods that can do this quite well like freeze casting with controlled temperature gradient (DOI: 10.1126/sciadv.1500849), magnetically assisted slip casting (<https://doi.org/10.1007/s11837-023-05705-w>), or even 3d printing using a large concentration of microplatelets (<https://doi.org/10.1038/s41598-017-14236-9>).

We recognize that this statement may have unintentionally given a lack of recognition for the advancements and capabilities of other state-of-the-art methods in replicating the complex hierarchical microstructures of natural materials. Our intention was to highlight the advantage of free-form capability of 3D printing, which enables the creation of diverse patterns according to specific designs.

In the revised manuscript, the introduction now more thoroughly explains the design rationale behind our own fabrication strategy, inspired by the hierarchical composite design principles observed in natural materials. We have removed the above comparisons between

our approach and other methods, as they were partial and unnecessary for our discussion. We hope that this revision will provide a clearer understanding of our work's value and more effectively highlight its significance and innovative aspects.

• Page 3, line 60: “novel fabrication strategy that achieves multi-scale coupling effects by integrating material processing techniques on the molecular and nano scale, and DIW 3D printing of hierarchical bioinspired architectures on the micro and macro scales” ->The claim of novelty here is inaccurate. There are other works published that do that already, for example: DOI: <https://doi.org/10.1038/s41586-018-0474-7>, DOI: <https://doi.org/10.1038/nmat4544>

We appreciate your comment and acknowledge that this claim regarding the novelty of our work may have been unspecific and inaccurate. As discussed in our responses above and in the revised manuscript, the novelty of this work lies in its design of a multi-scale fabrication strategy that implements the hierarchical composite design principles of natural materials in composite hydrogel engineering. We hope this refined focus justifies the significance of our research and differentiate its contribution from the cited literature. In addition, we have cited the above references in the introduction, acknowledging that they are important and relevant works on shear-induced alignment by DIW 3D printing.

• Page 3, line 63: the authors use ceramic microplatelets but in the introduction, they only mention the use of DIW to align microfibers along the printing direction. Because microplatelets are 2D and because some methods have been developed to create complex microstructures using DIW and controlled microplatelets orientations, these 3D printing methods should also be reviewed.

• Page 3, line 65: Concentric lamellar microstructures are also not new and may not enable the control over multiple architectures. Other papers that report on this: <https://doi.org/10.1038/s41598-017-14236-9>, DOI: [10.18063/ijb.v8i2.551](https://doi.org/10.18063/ijb.v8i2.551)

Thank you for the above comments and we have included the relevant works on shear-induced alignment by DIW 3D printing and concentric lamellar microstructures.

Recent advances in direct-ink-write (DIW) 3D printing also showcase its capability to align anisotropic particles along printed filaments by the extrusion shear force²⁵⁻²⁹.

25. Feilden, E. et al. *3D Printing Bioinspired Ceramic Composites*. *Sci Rep* **7**, 13759 (2017).
26. Sydney Gladman, A., Matsumoto, E. A., Nuzzo, R. G., Mahadevan, L. & Lewis, J. A. *Biomimetic 4D printing*. *Nature Mater* **15**, 413–418 (2016).
27. Gantenbein, S. et al. *Three-dimensional printing of hierarchical liquid-crystal-polymer structures*. *Nature* **561**, 226–230 (2018).
28. Dee, P., Tan, S. & Ferrand, H. L. *Fabrication of Microstructured Calcium Phosphate Ceramics Scaffolds by Material Extrusion-Based 3D Printing Approach*. *IJB* **8**, 551 (2022).
29. Li, T., Liu, Q., Qi, H. & Zhai, W. *Prestrain Programmable 4D Printing of Nanoceramic Composites with Bioinspired Microstructure*. *Small* **18**, 2204032 (2022).

• Page 4, line 69: ‘freeze-thawing twice’: since the authors are also using freezing, how is that less energy intensive as the other methods they were reporting on earlier?

We agree and appreciate your critical comment. Please see our previous response.

• Page 4, line 74: For the application for flexible electronics, do the final material contain water?

Yes, the final material retains water and thus it is characterized as an organo-hydrogel that features a water-glycerol binary solvent. Upon exposure to ambient conditions, the composite organo-hydrogel stabilizes after a minor mass loss due to water evaporation. Nonetheless, water does not completely evaporate at equilibrium, as evidenced by a mass loss of only ~12.5% (refer to Fig. 4c). This is due to the ideal miscibility of water and glycerol, alongside the hygroscopic characteristic of glycerol to attract and retain moisture from the environment. Therefore, the final composition contains both glycerol and water.

• Page 6, line 108: the ceramic content is 5 wt.%. It would be helpful to talk in terms of vol% to be able to compare with natural materials, which typically have between 60 and 96 vol% of mineral content, including the spicules the authors are referring to.

Thank you for the suggestion. Given an average density of 3.9 g/cm³ as specified by the manufacturer, the 5 wt.% ceramic content is equal to ~1.3 vol.%. This is indeed much less than the ceramic content of natural materials, which reflects the apparent contrast in the compositions of our composite organo-hydrogels and hard biological tissues. Again,

we wish to clarify that this study does not aim to exactly replicate the material compositions and structures of those natural prototypes, but rather leverage their hierarchical composite design and bioinspired architectures. Moreover, even with a relatively minimal ceramic reinforcement, our findings show significant mechanical enhancement in our composite organo-hydrogels, which indicates the effectiveness of our proposed fabrication strategy.

• Page 7, line 124: ‘excellent strength and toughness’: in comparison to what? It looks pretty low in comparison to metals, for example.

Thank you for your comment. This claim of “excellent strength and toughness” is made with respect to (composite) hydrogels. To avoid misinterpretation, we have revised the sentence as following:

It also revealed the highest work of extension (≈ 17.5 MJ/m³), indicating both its excellent strength and toughness among composite hydrogels.

• Page 7, line 135: “interlocked”. Given the extremely low concentration in ceramic microplatelets, is there any interlocking?

Thank you for raising this point. To answer this question, we have conducted CT scans for a clearer characterization of the microstructures of our materials. From the SEM and CT captures (Fig. 1 and Supplementary Fig. S5 and S7), certain interlocking of the ceramic platelets can be observed despite the low ceramic content. Nevertheless, we also recognize that this interlocking phenomenon may have been enhanced by samples’ drying shrinkage before the scans. In the original composite organo-hydrogels, such interlocking should be limited due to low concentrations of ceramic platelets. Accordingly, we have removed this point and revised our discussion to focus primarily on the main mechanism, which is the unidirectional alignment of ceramic platelets along the filament direction.

• Page 8, line 147: “cortical bones”. Cortical bones have more than 60 vol% mineral hence cannot compare with the current material.

Thank you for your comment. Here, our reference was aimed only at the concentric lamellar structure. As discussed previously, our revised manuscript now explicitly points out the difference in the compositions of our composite organo-hydrogels and the natural

prototypes, including cortical bones. Again, our bioinspired approach aims to implement their hierarchical composite design instead of exactly replicating their composition and structure.

• Figure 2, c, III: is the schematic adequate? Usually, 3D printers cannot print with control over the Z-direction. It does not correspond to the schematic in Figure 4, f.

Thank you for raising this concern. Yes, DIW 3D printing does not allow control over Z-direction, and therefore our bioinspired architectures are all designed and printed along the XY printing plane. However, Bouligand and crossed lamellar structures are designed with variations across different layers. In the Bouligand structure, the filaments are rotated at a constant 30° angle, while in the crossed lamellar structure as shown in Fig. 2c-III and its inset, the adjacent layers of zig-zag filaments interlock in a manner where the convex points of the zig-zag filaments in one layer align with the concave points in the layer above. These designed inter-layer variations have led to the crack deflection observed in Fig. 4f. In addition, we have included cross sectional SEM images of our materials with different architectures as attached below. We hope these clarifications help address your concerns.

Fig. S7. SEM images of the cross section of 3D printed composite organo-hydrogels with **a)** aligned, **b)** Bouligand, and **c)** crossed lamellar macro-architectures. Platelet orientations across the layers are marked in the right panel.

In the unidirectionally aligned sample, platelets are aligned into to the viewing plane. Both Bouligand and crossed lamellar samples showed varying platelet orientations: the Bouligand sample exhibits varying degrees of platelet orientation across the layers, while the crossed lamellar sample shows alternating platelet orientations in two adjacent layers.

• Figure 3: the property maps should include hydrogels with microplatelets and other microplatelet-based organic composites.

Thank you for your suggestion. Yes, we have compared our materials with (composite) hydrogels, including those with microplatelets such as graphene oxide (GO). Unfortunately, we have yet to find any relevant literature on hydrogel composites that uses similar ceramic platelets and conducts tensile testing comparable with ours. Meanwhile, we decide not to extend comparison to other microplatelet-based organic composites, given that polymers usually exhibit mechanical properties that are not directly comparable to those of hydrogels, due to their apparently differing nature. To keep our comparisons meaningful and relevant to the specific context of hydrogel-based materials, we prefer to focus on (composite) hydrogels.

• Page 18, line 315: How much water is there in the final composites? What was the humidity of the environment in which the relative mass loss tests were conducted?

Since glycerol is highly stable, we can assume that the relative loss of ~12.5 wt.% in the ambient environment is only attributed to water evaporation. At elevated temperatures, the relative mass loss increased to ~29 wt.%, which indicated the total water content. Thus, the final water content in the composite organo-hydrogels after reaching equilibrium can be estimated to be ~16.5 wt.%. The long-term stability test was conducted under the typical indoor environmental conditions of Singapore, where the humidity is usually 40 to 50%.

• Page 18, line 321: 0.97 W/mK for thermal conductivity is extremely low, how can it be suggested that it could help the thermal dissipation in flexible electronics? This is strong overclaim. The electrical conductivity of 7.1 S/m is also very low.

Thank you for your comment and we would like to clarify the context of our claims regarding thermal and electrical conductivity. Due to their large solvent content, hydrogels inherently have extremely low thermal conductivity, with that of our pure organo-hydrogel

being only $0.46 \text{ Wm}^{-1}\text{K}^{-1}$. Considering a small ceramic content, the increase in the thermal conductivity to $0.97 \text{ Wm}^{-1}\text{K}^{-1}$ (and up to $1.23 \text{ Wm}^{-1}\text{K}^{-1}$ at 10 wt.%) is notable. That said, we must clarify that our reference to facilitating heat dissipation in flexible electronics does not suggest using our composite organo-hydrogels as heat dissipating components. Rather, the increased thermal conductivity is intended to mitigate the thermal accumulation issue in flexible electronic applications, where the low thermal conductivity of conventional hydrogels could be a limitation.

Regarding electrical conductivity, the conductivity of 7.1 S/m for our materials is relatively high in the context of ion-conducting hydrogels, whose electrical conductivity is typically below 5 S/m. To address your concerns and avoid misinterpretations, we have revised our claims as below:

With a high solution content, pure organo-hydrogels had a very low thermal conductivity of $0.46 \text{ Wm}^{-1}\text{K}^{-1}$, which increased to $0.97 \text{ Wm}^{-1}\text{K}^{-1}$ at 5 wt.% ceramic content, and up to $1.23 \text{ Wm}^{-1}\text{K}^{-1}$ at 10 wt.% ceramic content (Supplementary Fig. S8). The enhanced thermal conductivity can help alleviate the heat buildup issue of hydrogels when they are used in flexible electronics (Supplementary Fig. S9).

With 2 wt.% FeCl_3 in the substituting solution, pure PVA organo-hydrogels revealed an electrical conductivity of 5.1 S/m, which further increased to 7.1 S/m in composite organo-hydrogels at only 5 wt.% ceramic content, and up to 8.8 S/m at 10 wt.% ceramic content (Supplementary Fig. S7). These values were higher than the conductivity of conventional ion-conducting hydrogels, which mostly falls below 5 S/m^{23} , and the improvement was achieved by only a small addition of ceramic platelets.

• Page 21, line 362: “novel class” -> please remove the claim of novelty.

Thank you for your suggestion. We have removed such claims of novelty.

Reviewer #3 (Remarks to the Author):

This paper reported a direct ink writing method to replicate the biological hierarchical structure. By extruding the composite hydrogel ink, the ceramic micro-platelets were aligned along the extrusion direction, and the concentric-circle-oriented structure was obtained. The organo-hydrogel was used to print composites with three bioinspired structural elements. The mechanical properties, such as stiffness, strength, and fracture energy, of the organo-hydrogels were enhanced by the multiscale energy dissipation mechanism. The composite prepared by the proposed method was extended to flexible electronics and validated for applications in robot-vehicle control and touch-screen interaction. However, major concerns listed below are needed to be clarified before this work is publishable.

1. The work is conceptually similar to Ref. 30 (Sci. Rep. 7, 13759 (2017)) or Adv. Mater. 35, 2211175 (2023). In these references, they also extruded composite inks containing ceramic platelets to obtain concentric circle-oriented lines and print the bioinspired Bouligand structure. They got the similar finding that bioinspired architectures increase energy absorption during fracture. Additional experiments and notes should be included to clarify what new conclusions have been developed compared to previous studies.

Thank you for highlighting the conceptual similarity of our work with these studies. Indeed, our work leverages DIW 3D printing and its shear-induced alignment to fabricate bioinspired hierarchical structures. However, the novelty of our study lies in its bioinspired multi-scale fabrication strategy, which not only leverages (1) filler alignment by DIW 3D printing but also encompasses (2) hydrogel matrix reinforcement by solution substitution and (3) ceramic-polymer interface treatment. Our approach is fundamentally inspired by the hierarchical composite design principles of natural composite materials, particularly hard biological tissues, which demonstrate a synergistic integration of (1) aligned stiff anisotropic particles or fibers, (2) embedded in a soft and tough matrix, (3) with a tight interface between the stiff elements and the soft matrix is observed. Accordingly, our fabrication strategy implements these bioinspired designs in composite organo-hydrogel filaments, which can be further utilized as fundamental building blocks for 3D printing various bioinspired hierarchical structures with their representative mechanical behaviors

and mechanisms. Therefore, our bioinspired approach represents a novel concept not yet explored or developed in the cited literature.

Also, we note that in most studies where a similar method is used, ceramic or ceramic composites with a large ceramic content are prepared to mimic hard biological tissues in nature, such as nacre. In contrast, our work implements the fundamental design principles of these hard biological tissues in our composite organo-hydrogels without mimicking their exact structure or composition. As a result, our findings demonstrate the applicability and effectiveness of these bioinspired design principles in soft engineering materials, which also provides a unique contribution to the understanding of bioinspired design principles and their mechanisms.

In response to your comment, we have thoroughly revised the abstract and introduction of our manuscript to delineate the concept of our research more effectively. We hope these revisions can address your concern and better highlight the novelty of our research.

2. In direct ink writing technique, the shear field induced by extrusion does not necessarily imply complete orientation of the platelets. Optimizing the printing parameters (flow rate, nozzle diameter, etc.) and ink composition ratios, as well as detailed characterization of the orientation degree of the platelets, would enhance the credibility of the article.

Thank you for your insightful comment. In our original manuscript, we briefly touched upon the process of shear-induced alignment because this method and its effectiveness in achieving platelet orientation have been well established in the existing literature, including our previous work (Small 18, 2204032 (2022)). More specifically, both our previous work and Sci. Rep. 7, 13759 (2017) underscore the use of long extrusion nozzles in promoting platelet alignment, which allows more time for platelets to align under shear forces during extrusion.

Still, we appreciate your recommendation for a more detailed characterization of platelet orientation. We have provided additional experiment and analysis in this revision by performing computed tomography (CT) scans on composite organo-hydrogel filaments extruded from both long cylindrical (with alignment) and short tapered nozzles (without alignment). This allowed us to directly observe and quantify the orientation of ceramic

platelets. By measuring the orientation angles of the platelets relative to the center of the filament cross-section, we provide a clearer structural characterization of filaments with aligned platelets in comparison to those with random platelets. The results below have been included in Fig. 1, and the relevant discussions are also provided here.

Through DIW 3D printing, previous studies showed that ceramic platelets would align by the extrusion shear force in the printing nozzle, and increasing the nozzle length could improve the platelet alignment^{25,29}. Fig. 1c, d compared the orientation distribution of ceramic platelets in filaments extruded through a short, tapered nozzle and a 35-mm long cylindrical nozzle, respectively, as determined from their computed tomography (CT) scans (Supplementary Fig. S3). Through the long cylindrical nozzle, a significant proportion of ceramic platelets were found in the lower angle range (0–10°), which indicates their effective alignment.

3. Fig. 2a(IV), b(IV), and c(IV) displayed SEM images where the layered interface might be a result of the printed lines, and the arrangement and distribution of ceramic platelets are still ambiguous. Finer 3D structural characterization of the printed hierarchical architectures will make the structure clearer.

Thank you for your recommendation to provide finer 3D structural characterization of our printed structures. In addressing this, we explored the use of CT scans but faced certain limitations due to a trade-off between the scan range and resolution. When scanning large samples with complete 3D printed architectures, the resolution achievable is insufficient to identify individual ceramic platelets and their orientations, considering their sub-micron thickness. On the other hand, while higher precision scans can visualize and differentiate

these platelets and their alignment, smaller samples used for this level of detail do not adequately represent the overall 3D printed architectures.

To navigate this limitation, we scanned smaller samples of 2x2 parallel filaments, which allowed us to observe platelet distribution and orientation in samples that consisted of multiple filaments, though not fully representative of the 3D printed architectures. We also supplemented this approach by including cross-sectional SEM images of samples with different bioinspired architectures. Given that our 3D printed bioinspired architectures are based on simple infill pattern designs, which is a straightforward concept, we hope these additional characterizations will help make the internal microstructures of our composite organo-hydrogels more comprehensible to readers and address the need for clarifying the hierarchical arrangement within our materials. Please see our revisions in the next two pages.

Fig. S5. CT scans of composite organo-hydrogel samples formed by 2x2 parallel filaments with **a-c)** aligned and **d-f)** randomly distributed ceramic platelets, including captures of transverse (middle) and longitudinal (right) views.

In the top panel (Fig. S5a-c), more pronounced platelet alignment can be observed in both transverse and longitudinal views. The scans also indicate good adhesion in between individual filaments. It should be noted that shape distortions are evident in these samples

due to drying. In particular, samples with aligned platelets demonstrate more significant distortions, which is also associated with their internal microstructure of aligned platelets.

Fig. S7. SEM images of the cross section of 3D printed composite organo-hydrogels with **a)** aligned, **b)** Bouligand, and **c)** crossed lamellar macro-architectures. Platelet orientations across the layers are marked in the right panel.

In the unidirectionally aligned sample, platelets are aligned into to the viewing plane. Both Bouligand and crossed lamellar samples showed varying platelet orientations: the Bouligand sample exhibits varying degrees of platelet orientation across the layers, while the crossed lamellar sample shows alternating platelet orientations in adjacent layers.

4. Fig.3d and Fig. 3e should include more mechanical gel composites, such as anisotropic PVA hydrogel (Nature 590, 594-599 (2021)), alginate hydrogels (Nat. Commun. 13, 3019 (2022)), PVA/aramid nanofiber hydrogel (Nat. Commun. 14, 759 (2023)), etc.

Thank you for your suggestion. We have thoroughly surveyed the literature to include composite hydrogels with duly reported tensile mechanical properties for our comparison. The PVA/PPy/ANF composite hydrogel in Nat. Commun. 14, 759 (2023) is a suitable addition into our comparison graphs (now Fig. 2g, h). As for the other suggested examples, we understand that they are essentially pure hydrogels reinforced by various sophisticated engineering methods, and their mechanical properties can significantly differ from our composite hydrogels and others in the literature. For your reference, we have performed such a comparison of pure hydrogels in a different work (10.1016/j.mattod.2023.11.014). Herein, we therefore decide to maintain a focused comparison among composite hydrogels (bulk and 3D printed) to keep our analysis relevant.

5. The authors emphasize that direct ink writing can produce hierarchical composites that exhibit superior mechanical properties. However, in terms of electronic devices, it is necessary to compare the performances with those of other architected electronic devices to highlight the advantages of the biomimetic hierarchical structures.

Thank you for your insightful comment. Overall, our work primarily focuses on demonstrating our bioinspired fabrication strategy and the potential applications of our composite organo-hydrogels, particularly in flexible electronics. Although we appreciate your suggestion to compare their performance with architected electronic devices, such comparisons can be affected by the vast diversity in material compositions, structural designs, and integration methods within different electronic devices. As such, we consider that this may have extended beyond the scope of our current study.

In this work, we have performed fundamental assessments of electrical conductivity, sensitivity, and cyclic sensing responses of our composite organo-hydrogels, along with the demonstration of a smart sensing glove to explore their application potential in flexible electronics. This also showcases the advantage of our approach that enables 3D printing of composite organo-hydrogels with varying mechanical properties and electrical sensitivities to meet different application needs.

As a final remark, we highly appreciate your suggestion and are motivated to further develop our materials for use in more complete, functional electronic devices, at which point we will perform detailed comparisons with the performances of existing devices as documented in the literature.

6. On Page 8, Line 157-158, "... demonstrated the precise material deposition of DIW 3D printing and excellent shape fidelity of the composite hydrogel ink." When discussing shape fidelity, quantitative data is needed to compare the discrepancies between the designed structure and the printed structure.

Thank you for your valuable suggestion. We have revised the original claim as below, supported with comparison between printed patterns and designed parameters to quantify the precision of DIW 3D printing with our composite inks.

Based on these bioinspirations, composite organo-hydrogels were 3D printed with infill patterns as designed in Fig. 2a-c II, through the precise material deposition of DIW 3D printing (Fig. 2a-c III and Supplementary Fig. S6).

Fig. S6. Precision of DIW 3D printing with composite ink across different bioinspired infill patterns. **a-c)** Optical images of printed patterns in aligned, Bouligand, and crossed lamellar samples, respectively. **d-f)** Comparison of the printed filament width, distance, and angle against the designed parameters, respectively.

REVIEWERS' COMMENTS

Reviewer #1 (Remarks to the Author):

The reviewer appreciated the authors for their efforts to address those concerns brought up by the three reviewers. However, all three reviewers have questioned the novelty of this work, and their response did not convince me the exact novelty that this work has delivered, since shear-induced alignment fillers has been widely reported, some representative examples have been listed by the reviewers. Therefore, I still cannot recommend the publication of this work in Nature Communications.

Reviewer #2 (Remarks to the Author):

I think the authors did a great job at revising their manuscript. I still have 2 comments:

1) Figure g and h are Ashby plot comparing the properties of hydrogel systems. This is good but it would be better to have the specific properties (ie, divided by the density) to compare them. Or select only composites that have the same concentration in reinforcement, for example. Or alternatively, mention in the text why the materials developed in this study are compared with these other composites. I am just worried that the materials here are better because there is a higher concentration of reinforcing particles.

2) My other comment is again relative to this bioinspiration. I still do not think that this work needs to be bioinspired to be interesting. So mostly, the bioinspiration features come from the 3D printing path. Could the authors comment how the properties they measure match their prediction from the bioinspired design, and if possible compare with other bioinspired and natural similar structures?

Reviewer #3 (Remarks to the Author):

The authors have well answered my questions and made the suitable revisions. Now, I can recommend it can be accepted as is.

Reviewer #2 (Remarks to the Author):

I think the authors did a great job at revising their manuscript. I still have 2 comments:

1. Figure 2g and h are Ashby plot comparing the properties of hydrogel systems. This is good but it would be better to have the specific properties (i.e., divided by the density) to compare them. Or select only composites that have the same concentration in reinforcement, for example. Or alternatively, mention in the text why the materials developed in this study are compared with these other composites. I am just worried that the materials here are better because there is a higher concentration of reinforcing particles.

Thank you for acknowledging the improvements made in our revised manuscript and for your further comments. We found density data for (composite) hydrogels often not reported in the literature, but to address your concern more directly, we took another review of relevant studies on composite hydrogels and included their filler content (in wt.%) in Supplementary Table S1. This addition indeed provides a more detailed basis for comparison. Additionally, we included two supplementary figures plotting tensile strength and work of extension against filler content, showing that our addition of 5 wt.% ceramic platelets are within the comparable range of those reported in other studies. We hope these updates provide a clearer comparative context and address your concerns.

Supplementary Fig. 8. Comparison of the composite organo-hydrogels in this work with other composite hydrogels in the literature in terms of **a**) tensile strength and **b**) work of extension against filler content. See Supplementary Table 1 for a detailed list of data.

“By incorporating both bioinspired structural and material engineering across multiple length scales, our hierarchical fabrication strategy achieves 3D printable strong and tough composite organo-hydrogels using a relatively small fraction of ceramic reinforcement (Supplementary Fig. 8). Therefore, it represents a promising approach to enhancing the mechanical properties of composite hydrogels while also introducing additional tunability in their mechanical responses.”

2. My other comment is again relative to this bioinspiration. I still do not think that this work needs to be bioinspired to be interesting. So mostly, the bioinspiration features come from the 3D printing path. Could the authors comment how the properties they measure match their prediction from the bioinspired design, and if possible compare with other bioinspired and natural similar structure.

Thank you for your continued attention to the bioinspired nature of our work, and the opportunity for us to clarify further the rationale behind printing our materials with different print paths. The essence of our bioinspired concept here is the combination of macro-scale architectural design and micro-scale reinforced unit filaments. Combining these two scales, we intend to mimic the “hierarchical” organization of natural materials, which is an essential component of our approach to applying the natural “hierarchical” composite design principles. Thus, the bioinspired concept does not solely reside in the print path designs, nor the concentric lamellar filaments, but rather in the hierarchical combination of the two, achieved seamlessly by the same DIW 3D printing process. To articulate this point more clearly, we have made the following updates to the manuscript:

“Leveraging the design freedom of DIW 3D printing, the composite organo-hydrogel filaments can be easily assembled as basic building blocks to construct free-form bioinspired architectures (Supplementary Fig. 5). This process combines micro-scale reinforced filaments with macro-scale architecture design, mimicking the hierarchical organization of natural materials. To demonstrate this, unidirectionally aligned, Bouligand, and crossed lamellar structures are selected by drawing inspiration from natural strong and tough materials (Fig. 2a-c).”

Regarding the matching mechanical properties of our materials and the natural prototypes, we have discussed on two mechanical behaviors: mechanical isotropy and fracture resistance. When first describing the natural prototypes, we have introduced their respective features: the unidirectionally aligned structure being anisotropic; the Bouligand structure featuring in-plane isotropy; and both the Bouligand and crossed lamellar structures known for high fracture toughness. Correspondingly, our materials match their natural prototypes as discussed in the excerpts below:

In terms of mechanical isotropy:

“Among them, unidirectionally aligned samples (Supplementary Fig. S7a) were the most anisotropic, as they exhibited the highest modulus and strength when stretched in the direction along the alignment of individual filaments, but saw significant reductions in strength, modulus, and strain in the normal direction. In the Bouligand samples, filaments were rotated at a constant $\theta = 30^\circ$ across the layers and thus not aligned in any particular direction (Fig. 2b, Supplementary Fig. S7b). This arrangement led to an in-plane mechanical isotropy without noticeable differences in their tensile properties when loaded in two normal directions. Featuring alternating curvatures in its zig-zag filaments (Fig. 2c, Supplementary Fig. S7c), the crossed lamellar architecture was also less anisotropic, with much smaller differences in its modulus and strain between two loading directions than those of the unidirectionally aligned. These results showed that the in-plane mechanical isotropy of composite organo-hydrogels with bioinspired macro-architectures closely resembled those of their natural prototypes.”

In terms of fracture resistance:

“The Bouligand architecture saw a further increased fracture energy of 26.1 kJ/m², where the rotating filaments led to effective crack pinning and crack deflection mechanisms that delayed and deflected crack propagation (Fig. 3b, f). This also increased the energy dissipation during crack propagation, which occurred along a more tortuous crack path as observed from the fracture surface (Fig. 3b and Supplementary Fig. S9). Remarkably, the crossed lamellar samples revealed the

highest fracture energy of 31.1 kJ/m². While some crack deflection was also observed (Fig. 3c), the crossed lamellar architecture led to a more prominent crack pinning effect with significantly delayed and slowed crack propagation (Fig. 3f). Due to the interlocking filament arrangement⁴¹, it could resist crack propagation while dissipating energy by stretching its zig-zag filaments, which significantly enhanced fracture toughness. Therefore, these results demonstrate that by 3D printing bioinspired macro-architectures, our strategy can translate the toughening mechanisms of natural materials into composite organo-hydrogels to effectively enhance their fracture toughness.”

Lastly, we appreciate your comment on comparing our work with other bioinspired and natural similar structures, which suggests potential directions for our future work. In the current study, our primary focus is to propose and investigate a bioinspiration concept of applying the natural hierarchical composite design principles in composite hydrogel fabrication. The aligned, Bouligand, and crossed lamellar structures, which are representative of the most common examples of natural materials, are selected mostly to demonstrate this concept. Certainly, building upon this work and further developing our approach by involving more intricate bioinspired architectures is an exciting prospect and we are keen to explore this in our future work. In response to your comment, we have included a brief discussion on potential directions for future research in the Discussion section of our manuscript.

“Hence, the proposed strategy effectively leveraged the hierarchical composite design principles of natural materials, achieving bioinspired mechanical mechanisms, and tunable mechanical and electrical sensing responses in the composite organo-hydrogels. The groundwork laid by this model strategy opens exciting opportunities for the development of advanced composite hydrogels. The versatility of DIW 3D printing in both material and structural design can be further exploited. Therein lies the potential to develop composite hydrogels with novel bioinspired architectures, further exploring the hierarchical organization of natural materials for enhanced performance.”